# A Multi-Input Convolutional Neural Network Model for Electric Motor Mechanical Fault Classification Using Multiple Image Transformation and Merging Methods

Insu Bae and Suan Lee *

School of Computer Science, Semyung University, Jecheon 27136, Republic of Korea; ibinsu@semyung.ac.kr
* Correspondence: suanlee@semyung.ac.kr; Tel.: +82-43-649-1273

**Abstract:** This paper addresses the critical issue of fault detection and prediction in electric motor machinery, a prevalent challenge in industrial applications. Faults in these machines, stemming from mechanical or electrical issues, often lead to performance degradation or malfunctions, manifesting as abnormal signals in vibrations or currents. Our research focuses on enhancing the accuracy of fault classification in electric motor facilities, employing innovative image transformation methods—recurrence plots (RPs), the Gramian angular summation field (GASF), and the Gramian angular difference field (GADF)—in conjunction with a multi-input convolutional neural network (CNN) model. We conducted comprehensive experiments using datasets encompassing four types of machinery components: bearings, belts, shafts, and rotors. The results reveal that our multi-input CNN model exhibits exceptional performance in fault classification across all machinery types, significantly outperforming traditional single-input models. This study not only demonstrates the efficacy of advanced image transformation techniques in fault detection but also underscores the potential of multi-input CNN models in industrial fault diagnosis, paving the way for more reliable and efficient monitoring of electric motor machinery.

**Keywords:** machine fault diagnosis; fault classification; electric motor machinery; deep learning; image transformation





## 1. Introduction

Machinery facilities play a crucial role in industrial production, and faults in such facilities can lead to production disruptions and increased costs. Therefore, research aimed at improving fault detection and prediction holds significant importance. The causes of electric motor faults are primarily classified into bearing, winding, environmental, and various other issues. Bearing faults mainly occur due to factors such as corrosion, insufficient lubrication, and wear. Winding faults are divided into electrical and mechanical causes, with electrical causes including overload, interturn short circuits, interphase short circuits, and momentary overvoltages, while mechanical causes involve issues like shaft constraints and direct contact between the rotor and stator. Environmental faults primarily arise from moisture corrosion and chemical substances in the surrounding environment.

Unpredictable faults can occur in various parts of the device due to aging or operating conditions, and if regular inspections and maintenance are delayed, serious problems can arise. When faults occur, they not only affect the operation of the electric motor but also have a negative impact on the entire system, including industrial processes, transportation, water supply and drainage, firefighting, and power systems. Therefore, technology that can predict and prevent faults in advance is necessary. Accurate lifespan prediction is crucial as inaccurate predictions can result in unforeseen costs. Industries such as railways, machinery, and electric motors face challenges in lifespan prediction due to various installation times, manufacturers, and specifications. When faults or performance degradation occur in electric motors, anomalies are typically observed in vibrations, currents, temperatures, etc., deviating from normal ranges. Analyzing faults and malfunctions in electric motors

requires practical experience, data collection, a comprehensive understanding of fault identification, and an understanding of causes and related symptoms.

The field of mechanical fault diagnosis has recently undergone substantial advancements, primarily driven by the application of deep learning techniques. Central to these developments is the use of convolutional neural networks (CNNs) for fault diagnosis, which has become increasingly prevalent [1]. This includes a range of approaches, from employing CNNs for fault classification in mechanical equipment [2] and small current grounded distribution systems [3] to using them for bearing fault classification, where methods integrate spectral kurtosis filtering and Mel-frequency cepstral coefficients [4], and even investigating the learning mechanisms of CNNs through time–frequency spectral images [5].

Additionally, novel network architectures and signal-processing techniques have been introduced to enhance fault diagnosis. This includes the development of a weight-sharing capsule network using one-dimensional CNNs [6], the exploration of one-dimensional local binary patterns (1D-LBPs) in bearing vibration signal analysis [7], and the use of one-dimensional ternary patterns (1D-TPs) for accurate fault diagnosis in bearings [8]. Advanced machine learning models like improved random forest algorithms have also been applied for industrial process fault classification [9] and rotating machinery fault diagnosis using multiscale dimensionless indicators [10].

Moreover, the field is witnessing a growing trend in employing complex machine learning strategies for fault detection. These include the application of kernelized support tensor machines (KSTMs) and multilinear principal component analysis (MPCA) for rotating machine fault detection [11], the use of ensemble machine learning-based fault classification schemes [12], and the integration of adaptive features extracted by modified neural networks for intelligent fault diagnosis [13]. Furthermore, cross-domain approaches and advanced deep learning strategies are gaining traction. This is evidenced by the use of tensor-aligned invariant subspace learning combined with 2DCNN for intelligent fault diagnosis [14], the implementation of deep transfer learning strategies for automated fault diagnosis [15], and the development of specialized models like the PrismPatNet for engine fault detection [16].

In addition, there is an increasing focus on using deep learning for more efficient and robust fault diagnosis under various conditions. Techniques like the deep focus parallel convolutional neural network (DFPCN) have been introduced to address imbalanced machine fault diagnosis [17], and simplified CNN structures have been proposed for more effective rolling bearing fault diagnosis [18]. Multisensor approaches using 2D deep learning frameworks are also being explored for distributed bearing fault detection [19]. Lastly, researchers in the field are exploring the potential of synthetic data generation using variational autoencoders for enhanced fault classification and localization in transmission networks [20].

To address these needs, this study proposes a method to improve fault detection and prediction, especially in electric motor installations in the industrial field. We utilized real mechanical equipment fault prediction sensor data [21], and in particular, we extracted 2.2 kW current data from subway station air conditioning equipment. In real industrial facilities, various noises may exist, and it may be difficult to use high-specification processing systems. Therefore, this study aims to improve the accuracy of fault classification by utilizing various multi-input CNN structures and image conversion techniques as a robust yet lightweight model. The contributions of this research are as follows:

- Automatically extracting and transforming key features of time series signals to use features in both time and frequency domains, and standardizing different formats such as sampling rate, duration, etc., through image transformation, so that the same model can be used in different datasets;
- Converting signals into images reduces the number of dimensions of the data and makes it easier to process efficiently in deep learning models using CNNs with a two-dimensional image representation;

- Through experiments, we compared different image conversion methods (RP, GASF, and GADF) and proposed a multi-input CNN structure that combines the conversion methods and shows more robust performance.

## 2. Materials and Methods

### 2.1. Dataset

The dataset [21] used in this study primarily encompassed the vibration and current data of mechanical devices installed at each station of the Daejeon Metro construction in South Korea. The main focus of the experiments was on a total of 41 motors installed in the air conditioning rooms of Daejeon Station, City Hall Station, and Gapcheon Station. The experiments emphasized data collection for five different operational states for each motor, including normal operation, bearing faults, rotor imbalance, shaft misalignment, and loosened belts. The data collection period spanned 4 months, with over a million data points for each state type.

Data collection involved measuring vibration and current signals using sensors and transmitting them to servers within the subway construction via an LTE-M network. The collected time series data were measured for 3 s at a base sample rate of 4 kHz, and the sample rate could be adjusted based on the device's condition. After verifying the integrity of the data, specified parameters were extracted and saved as CSV files.

The data structure depicted in Figure 1 consists of header and data sections. The header section is organized based on the characteristics of the mechanical device and data attributes. The data section includes actual collected data values such as time and acceleration. This data structure was designed to allow users to extract desired parameters using raw data.

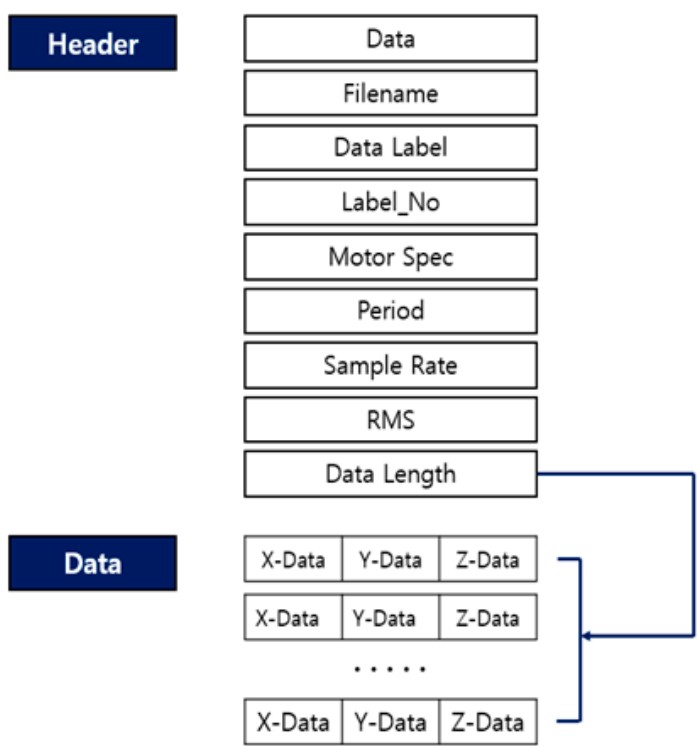

**Figure 1.** The structure of the motor mechanical fault data structure [21].

Machine facility data were broadly categorized into four types: bearing, belt, shaft, and rotor. Each data point was distinguished as normal or abnormal. In this paper, 2.2 kW motor data were utilized. Table 1 shows the annotation format for the dataset.

**Table 1.** Annotation formats for the datasets.

| Parameter | Type | Description |
|---|---|---|
| Date | string | Data collection date |
| Filename | string | Data filename |
| Data Label | string | Fault type |
| Label_No | string | Fault type unique number |
| Motor Spec | object [] | Motor rpm, rated power, rated current |
| Period | string | Collection time |
| Sample Rate | integer | Collected signal sample frequency |
| RMS | float | Effective value according to fault type |
| Data Length | integer | Data length |

### 2.2. Time Series Data

The objective of this paper was to leverage image encoding based on time series data for efficient classification. Various image encoding techniques were explored to transform time series data into 2D images. Figure 2 illustrates sample data before encoding. 'R', 'S', and 'T' represent the three phases commonly used in induction motor systems. Each phase is arranged at 120-degree intervals. The 'R phase' starts at 0 degrees, with both current and voltage increasing simultaneously. The 'S phase' starts 120 degrees later than the 'R phase' at 120 degrees, and finally, the 'T phase' starts 240 degrees later than the 'S phase'. These phases are used to generate and control the rotation of three-phase motors. Through various combinations of power and voltage, the direction and speed of the motor's rotation can be adjusted. The goal of this research was to enhance the efficiency of classification by utilizing various image encoding techniques and using the transformed images as inputs for deep learning models. We compared various encoding techniques to find the most effective approach in representing time series data in a format suitable for training deep learning models.

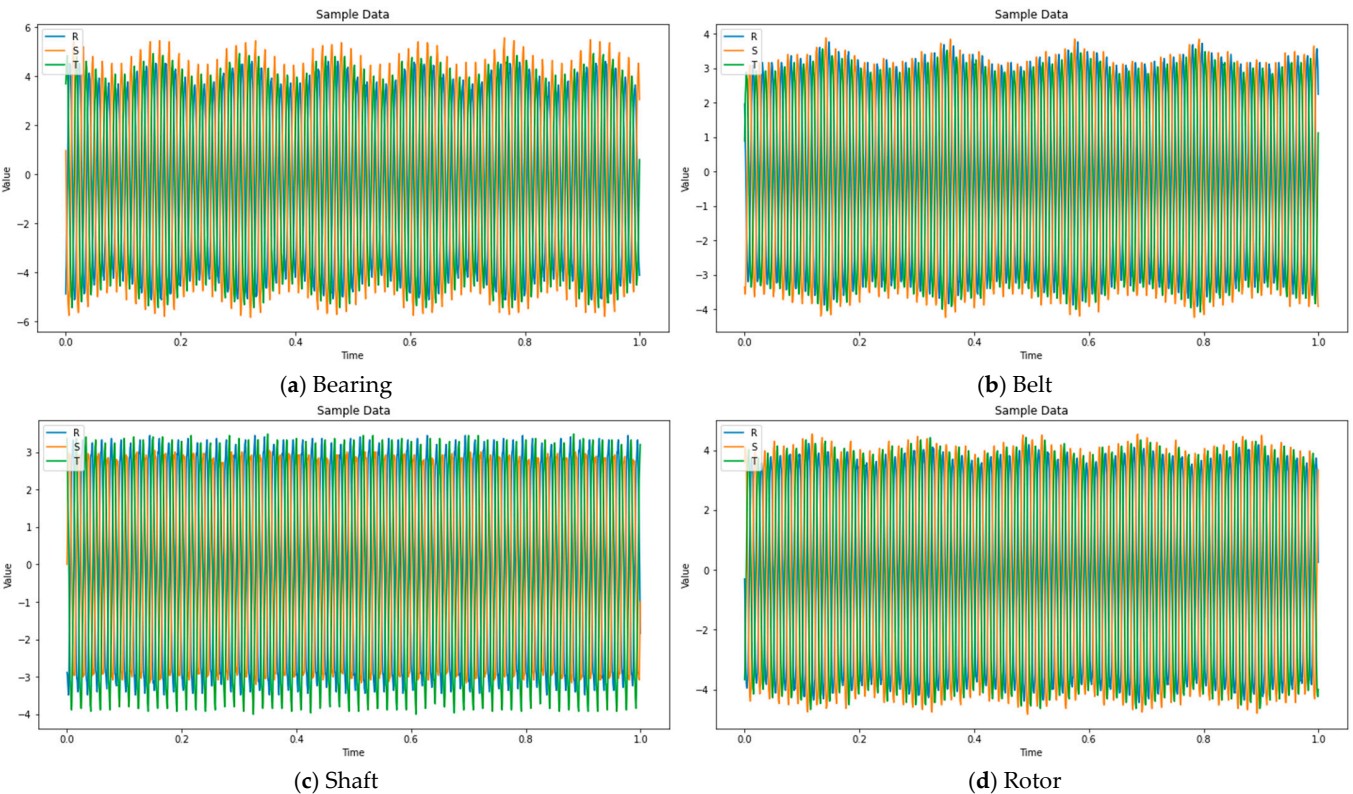

(**a**) Bearing

(**b**) Belt

(**c**) Shaft

(**d**) Rotor

**Figure 2.** Example of raw data from 'R', 'S', and 'T' phases for Bearing, Belt, Shaft, and Rotor.

*2.3. Feature Extraction*

In this study, latent encoding transformations of time series data were performed, enabling fault classification in a simple and lightweight model.

2.3.1. RP (Recurrence Plot)

The recurrence plot (RP) is a visualization technique used in the analysis of time series data [22]. They help uncover patterns, trends, and periodicities within the data by transforming them into two-dimensional representations. The primary goal of the algorithm is to explore the m-dimensional phase-space trajectories, as defined by Equation (1), where m represents the embedding dimension.

The m-dimensional phase-space trajectories are essentially representations of the underlying dynamics of the time series data. By examining these trajectories, one can gain insights into the system's behavior and identify recurrent patterns, which may indicate the presence of certain regularities or periodicities in the data.

The transformation of time series data into two-dimensional representations involves constructing a matrix known as the recurrence plot. This matrix is binary and reflects whether or not pairs of points in the time series are close to each other in the phase space. The binary nature of the matrix simplifies the complex temporal information into a more visually accessible form. The horizontal axis (X-axis) represents the passage of time, and the vertical axis (Y-axis) represents the similarity from one time step to the other: the closer the distance between points, the more similar the pattern at that time step. In Figure 3, the actual results of data transformation using the recurrence plot are depicted.

$$RP(i,j)\begin{cases} 1 \; if \left\| \vec{x}(i) - \vec{x}(j) \right\| \leq \epsilon \\ 0 \; otherwise, \end{cases} \tag{1}$$

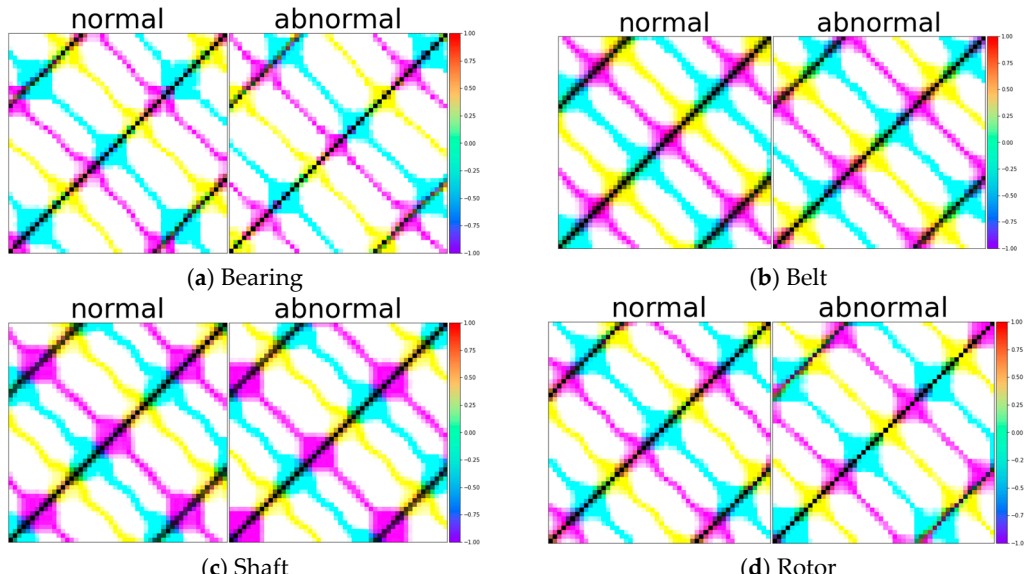

**Figure 3.** Samples of test data with RP (recurrence plot) for Bearing, Belt, Shaft, and Rotor.

2.3.2. GAF (Gramian Angular Field)

The Gramian angular field (GAF) is an algorithm that visually represents the temporal correlations in time series data using polar coordinates [23]. The matrix based on these polar coordinates has the advantage of more effectively preserving correlations when transforming time series data into images. The GAF is classified into two methods based on the sum and difference of angles. The Gramian angular summation field (GASF) is defined by the sum of angles in polar coordinate time series data, as shown in Equations (2) and (3). The horizontal axis (X-axis) represents the passage of time or a sequence of time series data,

and the vertical axis (Y-axis) represents the sum of the cosine values at each grid point after converting the data for that time step into degrees. This effectively represents the patterns and structure of time series data as an image. The results of the transformation of time series data using the $GASF$ are depicted in Figure 4.

$$GASF = \begin{bmatrix} \cos(\varnothing_1+\varnothing_1) \cdots \cos(\varnothing_1+\varnothing_n) \\ \cos(\varnothing_2+\varnothing_1) \cdots \cos(\varnothing_2+\varnothing_n) \\ \vdots \qquad \ddots \qquad \vdots \\ \cos(\varnothing_n+\varnothing_1) \cdots \cos(\varnothing_n+\varnothing_n) \end{bmatrix} \tag{2}$$

$$GASF = \overline{x} \cdot \overline{x} - \sqrt{1-\overline{x^2}} \cdot \sqrt{1-x^2} \tag{3}$$

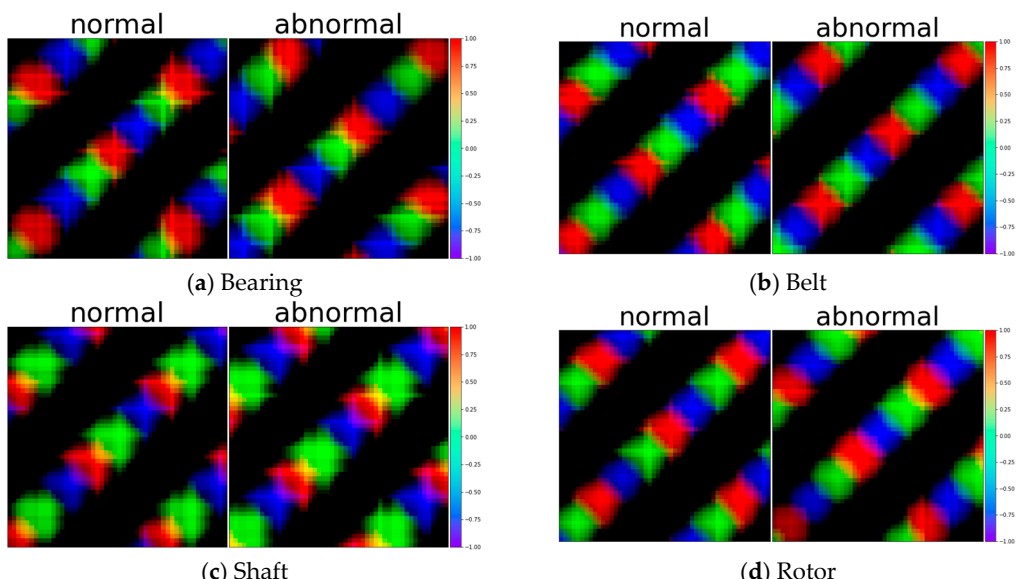

**Figure 4.** Samples of test data with GASF (Gramian angular summation field) for Bearing, Belt, Shaft, and Rotor.

On the other hand, the Gramian angular difference field (GADF) is defined by the difference of angles in polar coordinates, as shown in Equations (4) and (5). The GADF is similar to the GASF, but the vertical axis (Y-axis) represents the difference in cosine values after converting the data at that time step into degrees. This helps the GADF detect detailed data patterns and features by utilizing the difference in angles and is useful for capturing the different characteristics of time series data when used in conjunction with the GASF. The results of transforming a time series using the $GADF$ are illustrated in Figure 5.

$$GADF = \begin{bmatrix} \sin(\varnothing_1+\varnothing_1) \cdots \sin(\varnothing_1+\varnothing_n) \\ \sin(\varnothing_2+\varnothing_1) \cdots \sin(\varnothing_2+\varnothing_n) \\ \vdots \qquad \ddots \qquad \vdots \\ \sin(\varnothing_n+\varnothing_1) \cdots \sin(\varnothing_n+\varnothing_n) \end{bmatrix} \tag{4}$$

$$GADF = \sqrt{1-\overline{x^2}} \cdot \overline{x} - \overline{x} \cdot \sqrt{1-x^2} \tag{5}$$

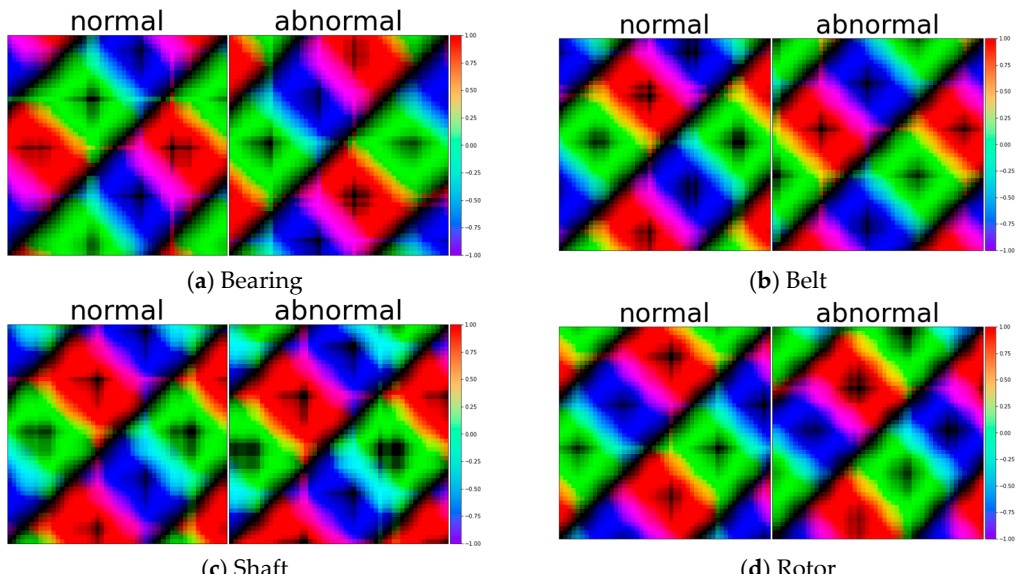

**Figure 5.** Samples of test data with GADF (Gramian angular difference field) for Bearing, Belt, Shaft, and Rotor.

## 3. Models

In this study, a convolutional neural network (CNN)-based model was employed for the classification of faults in electric motor machinery. This model was designed with a lightweight architecture, utilizing two convolutional layers and one max-pooling layer for feature extraction. For classification, three fully connected layers were integrated, and the output layer consisted of a single node. Binary classification was achieved through the sigmoid activation function. During the training process, the model was trained using binary cross-entropy loss function. A multi-input CNN was configured using encoded images from the RP, GASF, and GADF. Detailed explanations of the model architecture used in the experiments are provided in the following sections.

### 3.1. Single-Input CNN (Convolutional Neural Network) Model

The single-input CNN model described in Figure 6 and Table 2 was based on a lightweight architecture. This model took one of the RP, GASF, or GADF images as input and extracted features through two convolutional layers and one max-pooling layer. For classification, three fully connected layers were utilized, and binary classification was performed in the output layer with a single node. The rationale behind the selection of model hyperparameters involved using 32 filters of size $3 \times 3$ in the first convolutional layer to capture various features of the images. In the second convolutional layer, 64 filters were employed to extract more complex patterns. Dropout was applied at a 50% rate during training to prevent overfitting, thereby enhancing the model's generalization ability. The architecture of the model effectively reduced spatial dimensions by incorporating max-pooling layers after two convolutional layers. This allowed the model to capture both local and abstract features while maintaining computational efficiency. The fully connected layer with 256 neurons contributed to learning high-level abstract features and understanding intricate patterns.

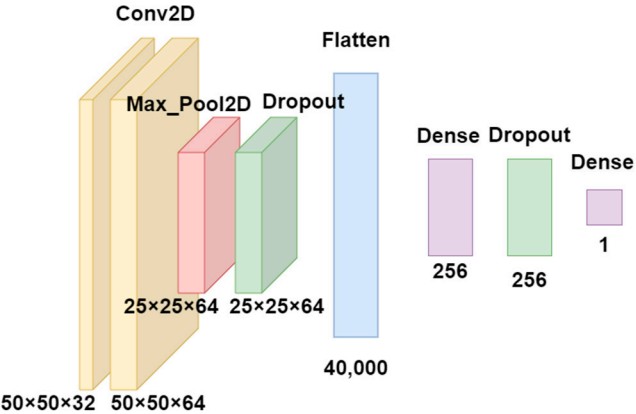

**Figure 6.** Single-input CNN model architecture.

**Table 2.** CNN (single input).

| Layer (Type) | Output Shape | Parameter |
|---|---|---|
| Conv2D | (50, 50, 32) | 896 |
| Conv2D | (50, 50, 64) | 18,496 |
| Max_Pool2D | (25, 25, 64) | 0 |
| Dropout | (25, 25, 64) | 0 |
| Flatten | (40,000) | 0 |
| Dense | (256) | 10,240,256 |
| Dropout | (256) | 0 |
| Dense | (1) | 257 |

### 3.2. Multi-Input CNN Model

The dual-input CNN model depicted in Figure 7 utilized input pairs such as {RP, GASF}, {RP, GADF}, and {GASF, GADF}. The features extracted from these image pairs were effectively combined using a merge layer, using various forms such as addition, concatenated, or average functions.

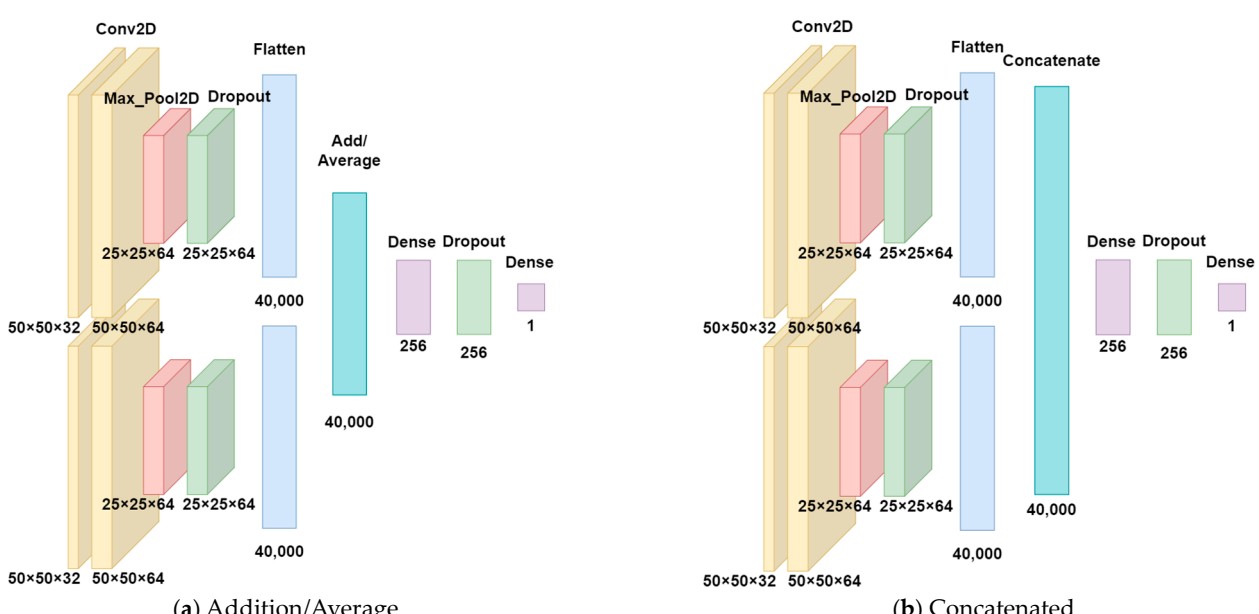

(**a**) Addition/Average  (**b**) Concatenated

**Figure 7.** Multi-input CNN model structure, where (**a**) two inputs are combined with an additive/average function, and (**b**) two inputs are combined with a concatenation function.

The "addition" layer takes multiple inputs and computes the element-wise sum of each input, generating a single output. This layer is commonly employed to process multiple inputs or combine outputs from specific layers. The concatenated layer takes multiple inputs and concatenates them, typically used to concatenate multiple inputs or combine outputs from specific layers. The average layer takes multiple inputs and computes the element-wise average based on all inputs at the same position, generating a single output. This layer is commonly used to average multiple inputs or average outputs from specific layers.

Both images were processed using convolution layers, max-pooling layers, and three fully connected layers for feature extraction and classification. The model performed binary classification, utilizing the sigmoid activation function and the binary cross-entropy loss function during training.

The triple-input CNN model described in Figure 8 utilized the {RP, GASF, GADF} image set as input. The structure of this model was similar to the dual-input CNN model but involved more inputs for feature extraction and classification.

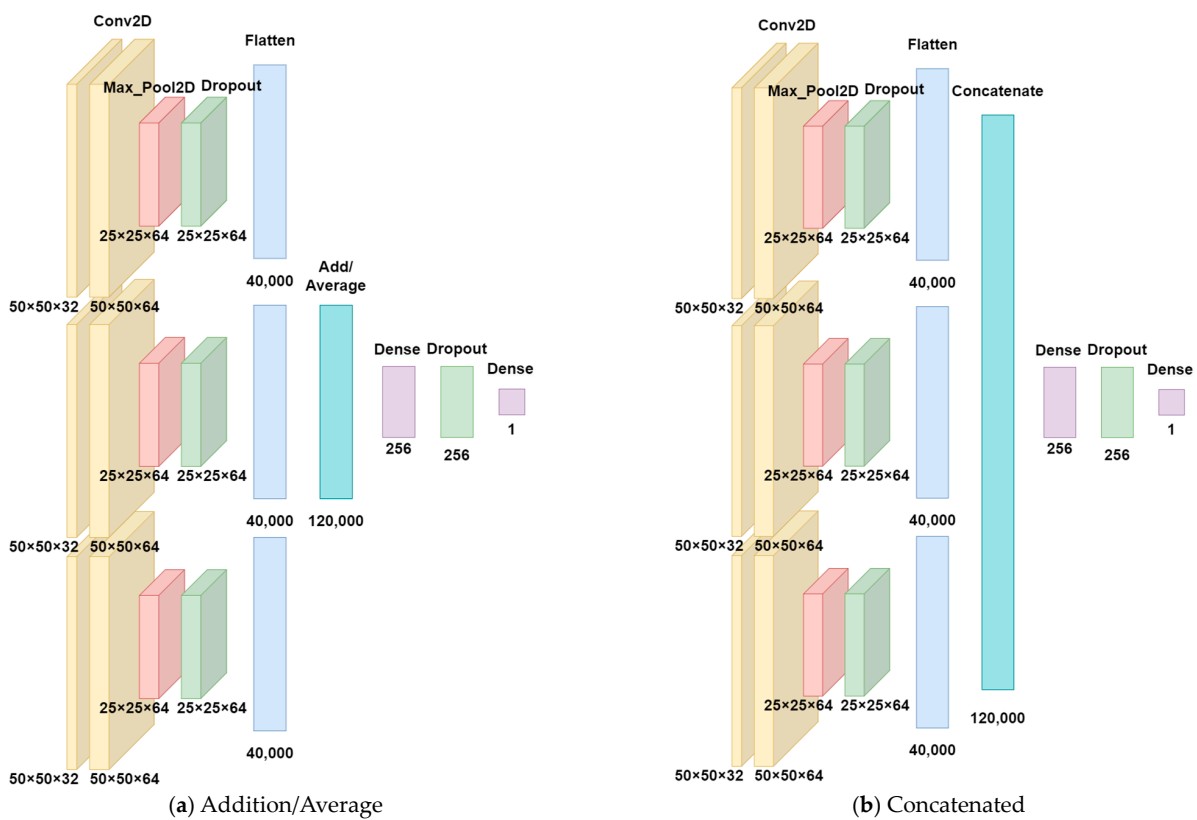

**Figure 8.** Triple-input CNN model architecture, where (**a**) three inputs are combined with an additive/average function, and (**b**) three inputs are combined with a concatenation function.

## 4. Experimental Procedures

### 4.1. Experimental Configuration

The belt dataset used in the experiment comprised 130,000 normal and 372,000 abnormal data, and the bearing dataset comprised 154,000 normal and 400,000 abnormal data. The shaft dataset encompassed 194,000 normal and 728,000 abnormal data, and the rotor dataset consisted of 1,334,000 normal and 458,000 abnormal data. For each dataset in the model, the ratio of train, test, and validate was 70:24:6. Experiments were conducted for each of the bearing, belt, shaft, and rotor datasets with 15 different configurations. These configurations included variations in input combinations (RP, GASF, and GADF) and model architectures (single-input CNN, concatenated multi-input CNN, and addition multi-input CNN). We also explored image fusion using single-input CNN.

For optimization, the Adam optimizer was chosen, and the initial learning rate was set to 0.001. The learning rate was fine-tuned through experimentation. Model performance was evaluated based on accuracy and binary cross-entropy loss. To evaluate the generalization performance of the model, we used early stopping and cross-validation.

### 4.2. Experimental Results

For performance evaluation, standard metrics such as accuracy, precision, recall, and F1 score were utilized. Confusion matrices were generated to examine the fault classification performance of each model in detail.

#### 4.2.1. Bearing

Through the comparison of the results in Table 3, it can be observed that the concatenate multi-input (GASF-GADF) and average multi-input (RP-GADF) had superior performance compared to other configurations. These models achieved 100% accuracy by correctly classifying all faults, and they also exhibited the highest precision, recall, and F1 scores. This indicates that models utilizing concatenation with multiple inputs achieved the most effective fault classification, particularly in the case of bearings. Figure 9 shows the loss and accuracy of the RP-GADF model in epochs, and Figure 10 shows the confusion matrix results.

**Table 3.** Model results for the bearing dataset.

| Input | Merging | Method | Accuracy | Precision | Recall | F1 Score |
|---|---|---|---|---|---|---|
| Single | None | RP | 0.999 | 1.000 | 0.997 | 0.998 |
| | | GASF | 0.999 | 1.000 | 0.998 | 0.999 |
| | | GADF | 0.999 | 1.000 | 0.998 | 0.999 |
| Multiple | Concatenated | RP-GASF-GADF | 0.995 | 0.993 | 1.000 | 0.997 |
| | | RP-GASF | 0.999 | 1.000 | 0.998 | 0.999 |
| | | **RP-GADF** | **1.000** | **1.000** | **1.000** | **1.000** |
| | | GASF-GADF | 0.999 | 1.000 | 0.999 | 0.999 |
| | Addition | RP-GASF-GADF | 0.999 | 0.998 | 1.000 | 0.999 |
| | | RP-GASF | 1.000 | 0.999 | 1.000 | 1.000 |
| | | **RP-GADF** | **1.000** | **1.000** | **1.000** | **1.000** |
| | | GASF-GADF | 0.997 | 1.000 | 0.996 | 0.998 |
| | Average | RP-GASF-GADF | 0.973 | 0.964 | 1.000 | 0.982 |
| | | RP-GASF | 0.999 | 0.999 | 1.000 | 0.999 |
| | | RP-GADF | 0.999 | 0.998 | 1.000 | 0.999 |
| | | **GASF-GADF** | **1.000** | **1.000** | **1.000** | **1.000** |

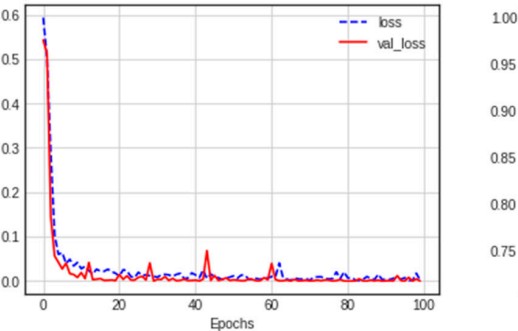 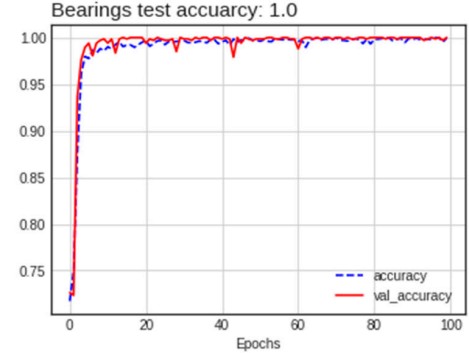

**Figure 9.** Loss and accuracy of the best model (concatenated RP-GADF) for the bearing dataset.

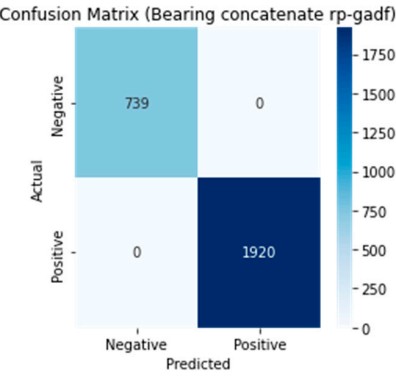

**Figure 10.** Confusion matrix of the best model (concatenated RP-GADF) for the bearing dataset.

4.2.2. Belt

As shown in the comparison results in Table 4, the concatenated multi-input (GASF-GADF) model exhibited superior performance compared to other configurations. This model had the best performance in terms of precision, recall, and F1 score. Figure 11 shows the loss and accuracy results of the model using the GASF-GADF method, and Figure 12 shows the confusion matrix results.

**Table 4.** Model results for the belt dataset.

| Input | Merging | Method | Accuracy | Precision | Recall | F1 Score |
|---|---|---|---|---|---|---|
| Single | None | RP | 0.740 | 1.000 | 0.740 | 0.851 |
| | | GASF | 0.997 | 0.995 | 0.996 | 0.995 |
| | | GADF | 0.993 | 0.996 | 0.977 | 0.987 |
| Multiple | Concatenated | RP-GASF-GADF | 0.741 | 0.741 | 1.000 | 0.851 |
| | | RP-GASF | 0.741 | 0.741 | 1.000 | 0.851 |
| | | RP-GADF | 0.741 | 0.741 | 1.000 | 0.851 |
| | | GASF-GADF | 0.997 | 0.997 | 0.999 | 0.998 |
| | Addition | RP-GASF-GADF | 0.741 | 0.741 | 1.000 | 0.851 |
| | | RP-GASF | 0.741 | 0.741 | 1.000 | 0.851 |
| | | RP-GADF | 0.741 | 0.741 | 1.000 | 0.851 |
| | | **GASF-GADF** | **0.998** | **0.999** | **0.999** | **0.999** |
| | Average | RP-GASF-GADF | 0.741 | 0.741 | 1.000 | 0.851 |
| | | RP-GASF | 0.741 | 0.741 | 1.000 | 0.851 |
| | | RP-GADF | 0.741 | 0.741 | 1.000 | 0.851 |
| | | GASF-GADF | 0.996 | 0.995 | 0.999 | 0.997 |

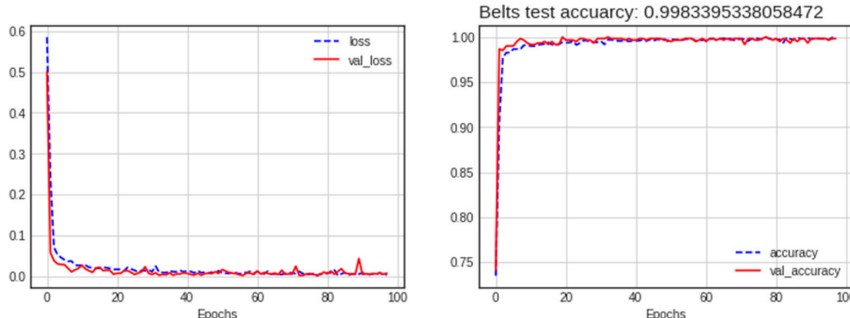

**Figure 11.** Loss and accuracy of the best model (addition GASF-GADF) for the belt dataset.

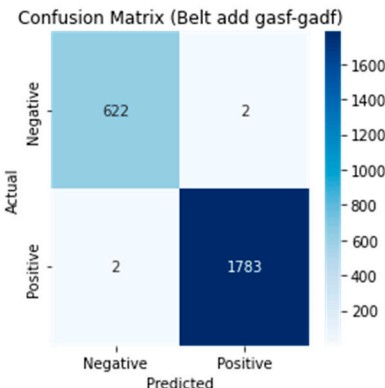

**Figure 12.** Confusion matrix of the best model (addition GASF-GADF) for the belt dataset.

### 4.2.3. Shaft

As shown in the comparison results in Table 5, the concatenated multi-input (GASF-GADF) model demonstrated superior performance compared to other configurations. This model exhibited the best performance in terms of accuracy, precision, and F1 score. In all evaluation metrics except recall, the model yielded excellent results in effectively predicting shaft faults. Figure 13 shows the loss and accuracy results in epochs for the best model, and Figure 14 shows the confusion matrix.

**Table 5.** Model results for the shaft dataset.

| Input | Merging | Method | Accuracy | Precision | Recall | F1 Score |
|---|---|---|---|---|---|---|
| Single | None | RP | 0.823 | 0.817 | 1.000 | 0.899 |
| | | GASF | 0.971 | 0.968 | 0.996 | 0.982 |
| | | GADF | 0.957 | 0.958 | 0.989 | 0.973 |
| Multiple | Concatenate | RP-GASF-GADF | 0.877 | 0.865 | 1.000 | 0.928 |
| | | RP-GASF | 0.896 | 0.884 | 1.000 | 0.938 |
| | | RP-GADF | 0.826 | 0.820 | 1.000 | 0.901 |
| | | **GASF-GADF** | **0.979** | **0.976** | **0.997** | **0.987** |
| | Addition | RP-GASF-GADF | 0.872 | 0.861 | 1.000 | 0.925 |
| | | RP-GASF | 0.968 | 0.963 | 0.998 | 0.980 |
| | | RP-GADF | 0.825 | 0.819 | 1.000 | 0.900 |
| | | GASF-GADF | 0.946 | 0.937 | 0.998 | 0.967 |
| | Average | RP-GASF-GADF | 0.907 | 0.895 | 1.000 | 0.945 |
| | | RP-GASF | 0.965 | 0.968 | 0.988 | 0.978 |
| | | RP-GADF | 0.827 | 0.820 | 1.000 | 0.901 |
| | | GASF-GADF | 0.973 | 0.976 | 0.989 | 0.983 |

### 4.2.4. Rotor

As shown in the comparison results in Table 6, the average multi-input (RP-GADF) model demonstrated superior performance compared to other configurations. This model achieved the highest accuracy and maintained top performance in precision, recall, and F1 score. Figure 15 shows the loss and accuracy results of the average RP-GADF method, and Figure 16 shows the confusion matrix results.

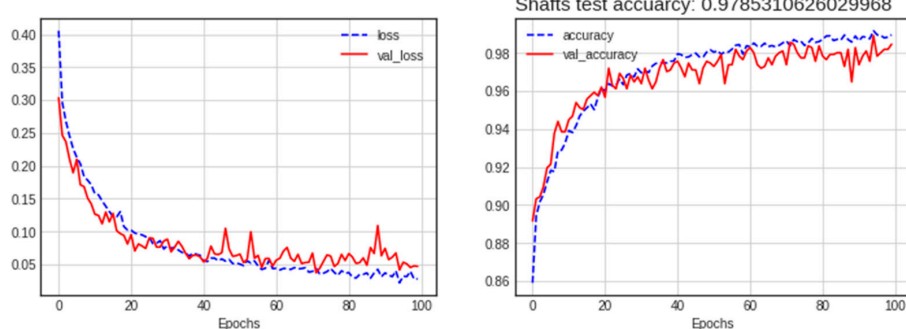

**Figure 13.** Loss and accuracy of the best model (concatenated GASF-GADF) for the shaft database.

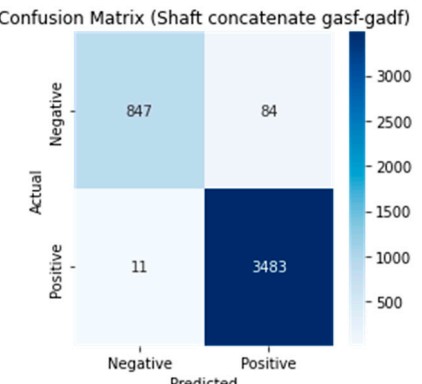

**Figure 14.** Confusion matrix of the best model (concatenated GASF-GADF) for the shaft database.

**Table 6.** Model results for the rotor dataset.

| Input | Merging | Method | Accuracy | Precision | Recall | F1 Score |
|---|---|---|---|---|---|---|
| Single | None | RP | 0.982 | 0.939 | 0.997 | 0.967 |
| | | GASF | 0.978 | 0.935 | 0.982 | 0.958 |
| | | GADF | 0.960 | 0.925 | 0.919 | 0.922 |
| Multiple | Concatenated | RP-GASF-GADF | 0.978 | 0.950 | 0.965 | 0.958 |
| | | RP-GASF | 0.978 | 0.942 | 0.975 | 0.958 |
| | | RP-GADF | 0.986 | 0.957 | 0.991 | 0.974 |
| | | GASF-GADF | 0.974 | 0.920 | 0.984 | 0.951 |
| | Addition | RP-GASF-GADF | 0.982 | 0.940 | 0.990 | 0.965 |
| | | RP-GASF | 0.969 | 0.898 | 0.992 | 0.942 |
| | | RP-GADF | 0.973 | 0.914 | 0.986 | 0.949 |
| | | GASF-GADF | 0.977 | 0.936 | 0.978 | 0.956 |
| | Average | RP-GASF-GADF | 0.985 | 0.958 | 0.983 | 0.970 |
| | | RP-GASF | 0.988 | 0.964 | 0.988 | 0.976 |
| | | **RP-GADF** | **0.992** | **0.981** | **0.989** | **0.985** |
| | | GASF-GADF | 0.949 | 0.917 | 0.882 | 0.899 |

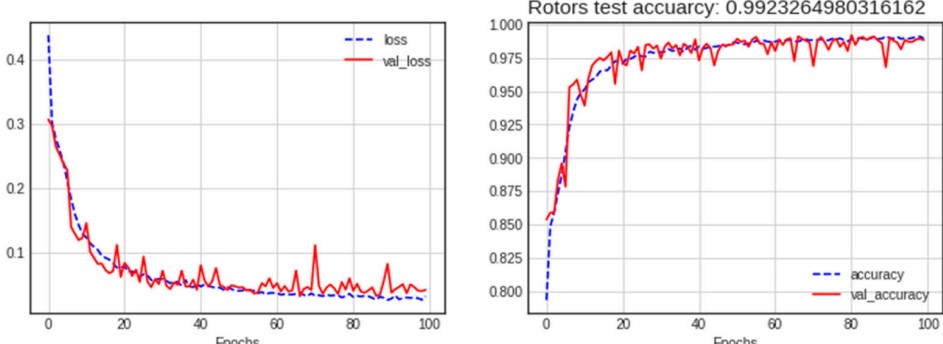

**Figure 15.** Loss and accuracy of the best model (average RP-GADF) for the rotor dataset.

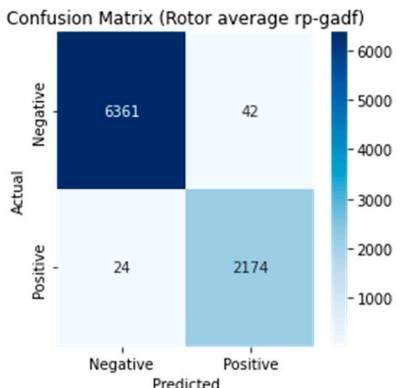

**Figure 16.** Confusion matrix of the best model (average RP-GADF) for the rotor dataset.

## 5. Discussion

In this study, we introduced a novel fault diagnosis model leveraging deep learning techniques, emphasizing feature extraction and image transformation to enhance performance. This model demonstrates significant potential in identifying and diagnosing faults in mechanical systems, offering a promising tool for predictive maintenance and operational efficiency.

However, the effectiveness of our proposed model is contingent upon the availability of substantial training data. This dependency poses a notable challenge, as acquiring a comprehensive dataset, particularly encompassing rare fault conditions or outliers, is inherently difficult in real-world scenarios. Such scarcity of fault data often leads to a class imbalance problem, which can skew the model's learning process and potentially compromise its diagnostic accuracy.

To address these challenges, we suggest a two-pronged approach. Firstly, the integration of simulation-based methods [24,25] into the fault diagnosis process presents a viable solution. By utilizing simulated data, we can artificially augment the dataset with a wider range of fault conditions, including those not commonly encountered in real-world operations. This approach not only helps in balancing the class distribution but also enriches the model's learning experience, potentially enhancing its diagnostic capabilities.

Secondly, the concept of continuous learning [26,27] in mechanical facilities and equipment is crucial. In dynamic industrial environments, where operating conditions and machine behaviors can evolve, the ability of a diagnostic model to adapt and learn continuously is paramount. This could be achieved through techniques like transfer learning, where a pretrained model is fine-tuned with new data, allowing it to adapt to new or changing fault patterns without the need for retraining from scratch.

Another critical aspect that warrants further research is the optimization of AI models for industrial applications. The current size and computational requirements of sophisticated deep learning models pose a challenge for their deployment in embedded systems

commonly used in industrial settings. Therefore, research focused on reducing the computational footprint of these models, without compromising their performance, is essential [28]. This could involve techniques like model pruning, quantization, or the development of more efficient neural network architectures.

In conclusion, while our proposed model shows promising results in fault diagnosis, its practical application is subject to overcoming challenges related to data availability, continuous learning, and model optimization for industrial deployment. Future research in these areas is not only necessary but will also significantly contribute to the advancement and practical utility of AI-driven fault diagnosis in the industrial sector.

## 6. Conclusions

This paper emphasizes the importance of motor failure detection and prediction, which can be utilized in various industries. Various types of faults that can occur in various mechanical facilities adversely affect industrial production, leading to production disruptions and increased costs, particularly in the industrial sector. We explored methods to enhance the accuracy of fault classification using a multi-input CNN structure and image transformation techniques. The experimental results showed that the proposed multi-input CNN model showed excellent results, and there were differences in the transformation and merging methods that showed excellent performance for each machine type (bearing, belt, shaft, rotor). This indicates that multi-input CNNs can be used as a more accurate and effective metric for machine fault detection in industrial environments compared to traditional methods.

Future research will include applying different transform methods such as Short-Time Fourier Transform (STFT) and Continuous Wavelet Transform (CWT) to improve the performance of the model. We will continue our efforts to improve the performance and generalization of the model by acquiring more and more diverse data.

**Author Contributions:** Conceptualization, S.L.; data curation, S.L.; formal analysis, I.B.; funding acquisition, S.L.; investigation, S.L.; methodology, S.L.; project administration, S.L.; resources, I.B.; software, I.B.; supervision, S.L.; validation, I.B. and S.L.; visualization, I.B.; writing—original draft preparation, I.B. and S.L.; writing—review and editing, S.L. All authors have read and agreed to the published version of the manuscript.

**Funding:** This paper was supported by Semyung University's University Innovation Support Project in 2023.

**Data Availability Statement:** The dataset can be downloaded at https://aihub.or.kr/aidata/30748 (accessed on 31 January 2024).

**Conflicts of Interest:** The authors declare no conflicts of interest.

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
