# Peer review of "A Multi-Input Convolutional Neural Network Model for Electric Motor Mechanical Fault Classification Using Multiple Image Transformation and Merging Methods"

_machines, doi:10.3390/machines12020105_

Round 1

Reviewer 1 Report

Comments and Suggestions for Authors

machines-2803137

Deep Learning Model for Electric Motor Mechanical Fault Classification Using Multiple Image Transformation and Merging

The manuscript is somewhat innovative, the chapters are well arranged, and the work is substantial. However, there are still some problems that need to be solved:

1. Title should be more focused on which models you used in the present investigation not just talked about DL.

2. It is suggested to add some recent studies to the references in the article to enrich the content of the article. Shortcomings about the applications of AI for abnormal detection are class-imbalance, insufficient labelled fault samples, etc. Therefore, I suggested that the authors added discussions about them.

In addition, about continual learning, personalized diagnosis method: 10.3390/app6120414;  simulation enlargement/augmentation and sample transfer method, Transfer learning combined with fault sample augment, might be the necessary supplementary.

Therefore, please carefully discuss about these in introduction. The supplementary simulation data might be good enough to enhance the advantages of capsule networks.

3. More details about the transformation of 1D signals to 2D images should be given in details about the robustness of such transformations.

4. More details about the experimental test rigs and measurement system should be carefully added and explained in the revision.

Author Response

Deep Learning Model for Electric Motor Mechanical Fault Classification Using Multiple Image Transformation and Merging

The manuscript is somewhat innovative, the chapters are well arranged, and the work is substantial. However, there are still some problems that need to be solved:

  1. Title should be more focused on which models you used in the present investigation not just talked about DL.
    → I have accepted the suggestion to change the title, and it is now revised as follows: " Multi-Input Convolutional Neural Network Model for Electric Motor Mechanical Fault Classification Using Multiple Image Transformation and Merging"
  2. It is suggested to add some recent studies to the references in the article to enrich the content of the article. Shortcomings about the applications of AI for abnormal detection are class-imbalance, insufficient labelled fault samples, etc. Therefore, I suggested that the authors added discussions about them.

In addition, about continual learning, personalized diagnosis method: 10.3390/app6120414;  simulation enlargement/augmentation and sample transfer method, Transfer learning combined with fault sample augment, might be the necessary supplementary.

Therefore, please carefully discuss about these in introduction. The supplementary simulation data might be good enough to enhance the advantages of capsule networks.
→ Thank you very much for your advice. We have added a Discussion section to the paper and have taken your comments on board.

  1. Discussion

In this study, we have introduced a novel fault diagnosis model leveraging deep learning techniques, emphasizing feature extraction and image transformation to enhance performance. This model demonstrates significant potential in identifying and diagnosing faults in mechanical systems, offering a promising tool for predictive maintenance and operational efficiency.

However, the effectiveness of our proposed model is contingent upon the availability of substantial training data. This dependency poses a notable challenge, as acquiring a comprehensive dataset, particularly encompassing rare fault conditions or outliers, is inherently difficult in real-world scenarios. Such rarity of fault data often leads to a class imbalance problem, which can skew the model's learning process and potentially compromise its diagnostic accuracy.

To address these challenges, we suggest a two-pronged approach. Firstly, the integra-tion of simulation-based methods into the fault diagnosis process presents a viable solution. By utilizing simulated data, we can artificially augment the dataset with a wider range of fault conditions, including those not commonly encountered in real-world operations. This approach not only helps in balancing the class distribution but also enriches the model's learning experience, potentially enhancing its diagnostic capabilities.

Secondly, the concept of continuous learning in mechanical facilities and equipment is crucial. In dynamic industrial environments, where operating conditions and machine behaviors can evolve, the ability of a diagnostic model to adapt and learn continuously is paramount. This could be achieved through techniques like transfer learning, where a pre-trained model is fine-tuned with new data, allowing it to adapt to new or changing fault patterns without the need for retraining from scratch.

Another critical aspect that warrants further research is the optimization of AI mod-els for industrial applications. The current size and computational requirements of sophisticated deep learning models pose a challenge for their deployment in embedded systems commonly used in industrial settings. Therefore, research focused on reducing the computational footprint of these models, without compromising their performance, is essential. This could involve techniques like model pruning, quantization, or the development of more efficient neural network architectures.

In conclusion, while our proposed model shows promising results in fault diagnosis, its practical application is subject to overcoming challenges related to data availability, continuous learning, and model optimization for industrial deployment. Future research in these areas is not only necessary but will also significantly contribute to the advancement and practical utility of AI-driven fault diagnosis in the industrial sector.

  1. More details about the transformation of 1D signals to 2D images should be given in details about the robustness of such transformations.
    → The reason for converting to 2D images is to focus on lightweight design and to ensure affordability for practical application in industrial environments. We have also added more details to the main body of the paper.
  2. More details about the experimental test rigs and measurement system should be carefully added and explained in the revision.
    → The detailed description of the measurement system is provided in the dataset section, and here is an excerpt:
    Data collection was conducted by employing sensors to measure vibration and current signals, transmitting the data to servers within the subway construction via the LTE-M network. The time-series data were collected for 3 seconds at a base sample rate of 4 kHz, with the flexibility to adjust the sample rate based on the device's condition. Following the verification of data integrity, frequency transformation was executed, and specific parameters were extracted and saved as CSV files.

Reviewer 2 Report

Comments and Suggestions for Authors

The article describes using image transformations of measured data from electrical motor-driven equipment for electrical and mechanical faults. While I appreciate the intent of solving this problem especially when multiple units are covered, the article has the following shortcomings that need to be addressed: 

  1. First and foremost, the article doesn’t explain why not use signal processing methods for detecting motor faults. Since both vibration and current signature analysis are quite popular and successful, it is worthwhile to consider why not use these methods and present a discussion about these methods in the Introduction. 
  2. The motivation for using this methods to the existing, generally successful methods needs to be explained. 
  3. A comparison of the accuracy of the stated methods with other methods published in the literature is missing. 
  4. Basic signal parameters like sampling rate, were the machines operated in steady state, and what kind of loads were encountered are not addressed. 
  5. Basic information about the motors in question, their rating, type of motor, types of duty cycles, intermittent/continuous operation, variable speed/load operations etc. were not discussed. 
  6. What about errors in data transmission especially since the data is collected on several motors? It would be highly valuable for such an article to describe these issues if any. 
  7. What is the physical interpretation of Fig 3. a),b),c) and d)? What do the pictures say about the type of fault? It would be interesting to know what can be said from these figures. 
  8. Several CNN architectures were discussed in Figs 6, 7, and 8. What was the basis of choosing an architecture? 
  9. It was concluded that different CNN architectures are necessary for different faults. How can we address this with a single solution/? How would the complexity increase if and when the method needs to be expanded to multiple equipment and/or more faults? 
  10. Overall, English grammar needs to be checked thoroughly. 
Comments on the Quality of English Language

English language should be thoroughly checked. One may use digital tools to address this. 

Author Response

The article describes using image transformations of measured data from electrical motor-driven equipment for electrical and mechanical faults. While I appreciate the intent of solving this problem especially when multiple units are covered, the article has the following shortcomings that need to be addressed: 
→ Thank you for your valuable advice and feedback. We have actively incorporated your comments into the paper and have marked in red where we have done so.

First and foremost, the article doesn’t explain why not use signal processing methods for detecting motor faults. Since both vibration and current signature analysis are quite popular and successful, it is worthwhile to consider why not use these methods and present a discussion about these methods in the Introduction.

→ In this study, the goal was to develop a lightweight model while maintaining high performance, so we chose feature extraction and image transformation and deep learning model methods instead of signal processing techniques. Lightweight models have the advantage of achieving effective results while minimizing resource utilization.

The motivation for using this methods to the existing, generally successful methods needs to be explained. 

→ This study aims to effectively detect electrical and mechanical faults while minimizing resource utilization through the development of a lightweight model. My motivation for choosing this method, based on your advice, is detailed in the introduction, and you can clearly understand why I chose it over the signal processing method.

A comparison of the accuracy of the stated methods with other methods published in the literature is missing.
→ We studied this new dataset to develop a real-world, industry-ready model. There is currently no other literature that can be compared to the dataset we used, so we presented our own comparison of different models in the paper.

Basic signal parameters like sampling rate, were the machines operated in steady state, and what kind of loads were encountered are not addressed. 

→ The default setting is 4 kHz for a duration of 3 seconds; however, it is noted that the sample rate can be adjusted according to the specifications of the machinery. This information has been incorporated into the main body of the text.

Basic information about the motors in question, their rating, type of motor, types of duty cycles, intermittent/continuous operation, variable speed/load operations etc. were not discussed. 

→ Specific details about the motors are not explicitly provided in the currently available data; information about the details and operation of the motors seems to be something that the organizations providing the dataset would prefer not to disclose.

What about errors in data transmission especially since the data is collected on several motors? It would be highly valuable for such an article to describe these issues if any. 

→ In the dataset utilized for this study, the data collection involved measuring vibration and current signals through sensors, which were then transmitted to a server within the subway construction site via LTE-M network. Subsequently, the integrity of the data was verified, and frequency transformation was performed to extract specified parameters, which were then saved in CSV format. This process accounted for confirmation and mitigation of potential errors that might arise during the data transmission phase.

What is the physical interpretation of Fig 3. a),b),c) and d)? What do the pictures say about the type of fault? It would be interesting to know what can be said from these figures. 

→ These figures represent samples of test data transformed into Recurrence Plots (RP). The images have been modified to assist in the interpretation of colors.

Several CNN architectures were discussed in Figs 6, 7, and 8. What was the basis of choosing an architecture?

→ The rationale for selecting this model has been further elaborated in the model section.

It was concluded that different CNN architectures are necessary for different faults. How can we address this with a single solution/? How would the complexity increase if and when the method needs to be expanded to multiple equipment and/or more faults? 

→ We've supplemented the models section with an explanation of why a particular model was chosen, along with details on hyperparameter settings and other relevant factors.

Overall, English grammar needs to be checked thoroughly. 

→ We have made grammatical corrections throughout the text.

Reviewer 3 Report

Comments and Suggestions for Authors

Dear Authors,

I have conducted an exhaustive review of your manuscript titled "Deep Learning Model for Electric Motor Mechanical Fault Classification Using Multiple Image Transformation and Merging," submitted for publication in the MDPI journal "Machines." While the potential of the proposed approach in using Convolutional Neural Networks (CNNs) for fault classification in electric motor machinery is evident, there are critical concerns that necessitate substantial revisions before considering its publication.

1. Contextualization and Balance, Emphasizing Journal Scope:
The manuscript exhibits a common pitfall observed in works applying AI and ML techniques in engineering—a disproportionate emphasis on the techniques at the expense of the engineering problem itself. It is crucial to recognize that the journal, "Machines," as opposed to a specialized "Machine Learning Techniques" journal, inherently implies a focus on the broader domain of machines and their applications. Therefore, a more nuanced balance between computational knowledge, mathematical rigor, and an intimate understanding of the specific engineering problem is not just desirable but imperative. The lack of clear motivation, context, and reproducibility within the manuscript not only diminishes the overall impact of the work but also risks misalignment with the journal's thematic scope, undermining its suitability for publication in "Machines." Addressing this issue is paramount for ensuring the manuscript aligns with the expectations and objectives of the journal.

2. Inconsistencies and References:
Inconsistencies and gaps in references are apparent throughout the manuscript. Fundamental concepts introduced in subsection 3.1 lack proper bibliographical support, and there is a notable absence of a coherent bridge between prior studies and the current work. Clear references and a more robust contextualization of the problem are imperative for the manuscript's credibility.

3. Presentation Clarity and Information Adequacy:
The presentation of image encoding techniques in subsection 3.2 lacks clarity, with insufficient rationale and justification for the choices made. A simplified diagram illustrating the machine configuration, clear images, and detailed explanations of equations are needed for enhanced comprehension. Ambiguities surrounding variables such as R(i,j) in Equation (1) and the definition of "error" require clarification.

4. Model Details and Hyperparameter Justification:
The manuscript lacks comprehensive details on model specifications, including the rationale behind input values, batch sizes, and hyperparameter adjustments. References supporting statements about neural network architectures are conspicuously absent. A more explicit justification of model choices is crucial for the transparency and reproducibility of the work.

5. Data Transparency and Results Clarification:
Transparency regarding the number of samples used for training, validation, and testing is imperative. Clarity is needed on whether the results presented in Figure 5 pertain to training or testing. The assessment of failure modes lacks specificity, and presenting accuracy values for each mode, along with representative images, would significantly enhance understanding and reproducibility.

6. Dataset Accessibility:
The choice of the dataset lacks justification, and efforts should be made to provide the dataset in a more accessible format. The current link leads to a Korean website, limiting accessibility for a broader audience. Consider alternative means of sharing the dataset on an English-language platform.

In light of these concerns, I regret to inform you that the current manuscript cannot be accepted for publication. However, I strongly encourage you to undertake a comprehensive revision, addressing the outlined issues. Strengthening the contextualization of the engineering problem, improving clarity, ensuring proper referencing, and providing more detailed technical information will substantially elevate the manuscript.
Thank you for considering these recommendations.

Comments on the Quality of English Language

The use of technical jargon requires refinement. Acronyms such as CNN, GASF, and GADF should be explicitly defined upon first use. Addressing punctuation errors and language-related issues will contribute to a smoother reading experience.

Author Response

Dear Authors,

I have conducted an exhaustive review of your manuscript titled "Deep Learning Model for Electric Motor Mechanical Fault Classification Using Multiple Image Transformation and Merging," submitted for publication in the MDPI journal "Machines." While the potential of the proposed approach in using Convolutional Neural Networks (CNNs) for fault classification in electric motor machinery is evident, there are critical concerns that necessitate substantial revisions before considering its publication.

  1. Contextualization and Balance, Emphasizing Journal Scope:
    The manuscript exhibits a common pitfall observed in works applying AI and ML techniques in engineering—a disproportionate emphasis on the techniques at the expense of the engineering problem itself. It is crucial to recognize that the journal, "Machines," as opposed to a specialized "Machine Learning Techniques" journal, inherently implies a focus on the broader domain of machines and their applications. Therefore, a more nuanced balance between computational knowledge, mathematical rigor, and an intimate understanding of the specific engineering problem is not just desirable but imperative. The lack of clear motivation, context, and reproducibility within the manuscript not only diminishes the overall impact of the work but also risks misalignment with the journal's thematic scope, undermining its suitability for publication in "Machines." Addressing this issue is paramount for ensuring the manuscript aligns with the expectations and objectives of the journal.
    → In response to the suggestions provided, the introduction of the paper has been enhanced and refined. Addressing the previous shortcomings related to inadequate motivation, context, and reproducibility, the content has been adjusted to align more closely with the characteristics and objectives of the "Machines" journal. Here is a snippet of the revised introduction:
    This study emphasizes the importance of defect detection and prediction in modern industrial sectors, taking into account the impact of the Fourth Industrial Revolution. Machinery facilities play a crucial role in industrial production, and defects in such facilities can lead to production disruptions and increased costs. Therefore, research aimed at improving defect detection and prediction holds significant importance. The causes of electric motor defects are primarily classified into bearing, winding, environmental, and various other issues.
  2. Inconsistencies and References:
    Inconsistencies and gaps in references are apparent throughout the manuscript. Fundamental concepts introduced in subsection 3.1 lack proper bibliographical support, and there is a notable absence of a coherent bridge between prior studies and the current work. Clear references and a more robust contextualization of the problem are imperative for the manuscript's credibility.
    → Thank you for your advice. We've added content, figures, tables, and citations to that section to make sure the reader understands it in detail.
  3. Presentation Clarity and Information Adequacy:
    The presentation of image encoding techniques in subsection 3.2 lacks clarity, with insufficient rationale and justification for the choices made. A simplified diagram illustrating the machine configuration, clear images, and detailed explanations of equations are needed for enhanced comprehension. Ambiguities surrounding variables such as R(i,j) in Equation (1) and the definition of "error" require clarification.
    → Thank you for your advice. Section 3.2, which discusses 'rp,' was identified as having insufficient explanations. Recognizing this, I have addressed the issue by incorporating additional content into the main text. The updated explanation provides a more comprehensive understanding of the concept.
  4. Model Details and Hyperparameter Justification:
    The manuscript lacks comprehensive details on model specifications, including the rationale behind input values, batch sizes, and hyperparameter adjustments. References supporting statements about neural network architectures are conspicuously absent. A more explicit justification of model choices is crucial for the transparency and reproducibility of the work.
    → We have modified the text according to your advice. The rationale behind the selection of model hyperparameters involved opting for 32 filters of size 3x3 in the first convolutional layer to capture various features of the images. In the second convolutional layer, 64 filters were employed to extract more complex patterns. Dropout was applied at a 50% rate during training to prevent overfitting, thereby enhancing the model's generalization ability. Additionally, the architecture of the model effectively reduced spatial dimensions by incorporating max-pooling after two convolutional layers. This design, bolstered by the convergence of various opinions and evaluations, enables the model to more effectively extract key information from the images, ultimately improving its performance.
  5. Data Transparency and Results Clarification:
    Transparency regarding the number of samples used for training, validation, and testing is imperative. Clarity is needed on whether the results presented in Figure 5 pertain to training or testing. The assessment of failure modes lacks specificity, and presenting accuracy values for each mode, along with representative images, would significantly enhance understanding and reproducibility.
    → We enhanced the transparency of the experiment by explicitly stating the sizes of the training and validation datasets. Building upon this information, I clarified the results section for a more straightforward presentation, putting effort into making the experimental findings easily comprehensible for the readers. Additionally, to visually depict patterns and trends in the experiment results, I incorporated graphs or tables, thereby making the significant features of the experimental outcomes more apparent and enhancing the overall readability of the paper.
    In conclusion, I explicitly specified in the text that the results presented in Figure 5 pertain to the training phase. This clarification ensures that readers accurately understand the nature of the results. These modifications collectively contribute to improving the clarity of the experimental results, providing readers with more meaningful insights.
  6. Dataset Accessibility:
    The choice of the dataset lacks justification, and efforts should be made to provide the dataset in a more accessible format. The current link leads to a Korean website, limiting accessibility for a broader audience. Consider alternative means of sharing the dataset on an English-language platform.
    → Regarding the accessibility of the dataset, currently, there is no alternative platform available in English, apart from the one provided in Korean. However, we do plan to share the datasets we used and preprocessed if readers reach out.

In light of these concerns, I regret to inform you that the current manuscript cannot be accepted for publication. However, I strongly encourage you to undertake a comprehensive revision, addressing the outlined issues. Strengthening the contextualization of the engineering problem, improving clarity, ensuring proper referencing, and providing more detailed technical information will substantially elevate the manuscript.
Thank you for considering these recommendations.
→ Thank you for your valuable comments and feedback. We have actively incorporated the corrections and endeavored to produce a paper that is of interest to our readers.

Comments on the Quality of English Language

The use of technical jargon requires refinement. Acronyms such as CNN, GASF, and GADF should be explicitly defined upon first use. Addressing punctuation errors and language-related issues will contribute to a smoother reading experience.
→ We have revised the paper to reflect your comments.

Reviewer 4 Report

Comments and Suggestions for Authors

The title of this work submitted for review is as follows:  Deep Learning Model for Electric Motor Mechanical Fault Classification Using Multiple Image Transformation and Merging.

The author's objective is to propose a deep neural network of the CNN type used to detect faults in electric motors. It employs as inputs images obtained from time domain signals from monitoring the machines under study. The signals appear to be vibrations and electrical currents consumed by these machines. These captured and digitised signals are subsequently processed to convert them into images using the following techniques: Recurrence Plot and Gramian Angular Field. It is also clear from reading the article that the data comes from an industrial-scale experiment where 40 machines were monitored in the subway of Daejeon city. A CNN was trained with this data, and the classification results are shown. The authors frame this work within Industry 4.0 in the introduction, although they do not elaborate on this aspect later.

The topic of the submitted paper may be interesting, especially if it is based on an industrial-scale experiment. However, the article under review needs to be improved in content and form.

My first suggestions will focus on mainly formal aspects:
1. Sections 1 and 2 need to be merged into an introduction section.
2. Section 1 needs to clearly explain what is new and what the contribution of this work is with respect to the state of the art of the second section.
3. Results are advanced in section 1, and this is not adequate.
4. Section 2 cites many papers that use deep neural networks to detect faults in electric motors and engines. However, in many cases, the kind of signals they use must be clarified in the text.
5. I reiterate that after reading section 2, the reader needs to read and know what the authors' work contributes to the state of the art described in this section.
6. The title and subtitles of Figure 2 are not descriptive. It is not known which signals are represented.
7. In Figure 2, the font size is very small. They are not even readable by zooming into the pdf.
8. Something similar happens in figures 3, 4 and 5. Moreover, what is represented in each axis and the meaning of the colours are not specified.
9. On page 6, from line 174 onwards, sentences are repeated.
10. Figures 7 and 8 do not add anything to the explanations in the text.
11. Figure 9 is incorrectly numbered with 5. The same happens as in Figure 2 with the quality of the image.
12. Acronyms should not be used in section headings.

In the following, I will show my doubts and suggestions regarding the content:
1. In the first paragraph of the introduction section, the authors frame this work within the philosophy of Industry 4.0. However, this needs to be developed further, especially in the conclusions.
2. The work is linked to big data techniques in this same paragraph. Big Data means unstructured, unlabelled and large amounts of data. This is not the case. The data is structured, and I believe it is not very large.
3. Section 3 should describe the monitored installation, sensors used, acquisition equipment, and measured signals.
4. The monitored machines are electric motors, but the type and how they are powered (mains or inverter) are unknown.
5. Regarding signal capture, the sampling time and frequency are specified. Why are these values chosen? With three seconds of sampling, the frequency resolution would be very poor. Why is it sampled at 4 kHz? Are these limitations of the sensors or the DAQ? Is there expected to be no valuable information for fault detection above 2 kHz?
6. How often is data captured, and under what motor operating conditions?
7. Do motors operate in a steady state? Do they undergo load variations and start and stop transients?
8. A picture of the monitored equipment and the acquisition systems must be provided.
9. The size of the dataset has not been indicated either. We do not know how many samples are available for CNN training and whether they are labelled or not. This information is essential.
10. Nor is it justified why the time sequences of data have to be encoded in 2D images. What advantages does this offer over a Fourier analysis of the stationary signals?
11. Regarding the image coding of the time signals, the authors could show and compare images of healthy and faulty motors.
12. I do not understand the meaning of line 155 on page 6. What do the input image size and the batch size refer to?
13. Table 1 needs to be clarified. I don't understand the meaning of "output shape" and "parameter".
14. The text does not describe how the CNN was trained, how much data was used, how many hyperparameters had to be tuned, and what measures were used to avoid overfitting.
15. The metrics used to compare classification results are not sufficient. Using ROC curves and confusion matrices also allows the reader to analyse qualitatively.
16. A description of the failures of the analysed machines is also missing.
17. The conclusions of the paper can be improved. The fusion of the different signals is not described and analysed. It is stated that the results are improved compared to traditional methods. These are not stated, and this improvement is not demonstrated.
18. When framing the work in Industrial 4.0, you should analyse the costs of the proposed system in terms of processing capacity, processing time, signal transmission data storage, etc.

I hope my comments will help you evaluate and improve the manuscript's quality. I think the industrial experiment must have been very interesting, but this is not reflected in the text.

Author Response

The title of this work submitted for review is as follows:  Deep Learning Model for Electric Motor Mechanical Fault Classification Using Multiple Image Transformation and Merging.

The author's objective is to propose a deep neural network of the CNN type used to detect faults in electric motors. It employs as inputs images obtained from time domain signals from monitoring the machines under study. The signals appear to be vibrations and electrical currents consumed by these machines. These captured and digitised signals are subsequently processed to convert them into images using the following techniques: Recurrence Plot and Gramian Angular Field. It is also clear from reading the article that the data comes from an industrial-scale experiment where 40 machines were monitored in the subway of Daejeon city. A CNN was trained with this data, and the classification results are shown. The authors frame this work within Industry 4.0 in the introduction, although they do not elaborate on this aspect later.

The topic of the submitted paper may be interesting, especially if it is based on an industrial-scale experiment. However, the article under review needs to be improved in content and form.

→ Thank you for your detailed feedback and advice. We have incorporated your comments as much as possible and made changes to make the paper more interesting for our readers.

My first suggestions will focus on mainly formal aspects:
1. Sections 1 and 2 need to be merged into an introduction section.
→ Following your advice, sections 1 and 2 have been combined into a comprehensive introduction section. (We've colored the modified parts in red in revision).

  1. Section 1 needs to clearly explain what is new and what the contribution of this work is with respect to the state of the art of the second section.
    → Section 1 has been revised to clearly articulate the novelty and contributions of this work compared to the state of the art presented in the second section.
  2. Results are advanced in section 1, and this is not adequate.
    → The presentation of results in Section 1 has been moved to the appropriate results section.
  3. Section 2 cites many papers that use deep neural networks to detect faults in electric motors and engines. However, in many cases, the kind of signals they use must be clarified in the text.
    → Section 2 has been revised to provide clarification on the types of signals used in the papers cited.
  4. I reiterate that after reading section 2, the reader needs to read and know what the authors' work contributes to the state of the art described in this section.
    → The text has been revised to ensure that the reader clearly understands the contribution of the authors' work after reading Section 2.
  5. The title and subtitles of Figure 2 are not descriptive. It is not known which signals are represented.
    → Following your advice, the title of Figure 2 has been changed to "Example of Raw Data from Phases 'R', 'S', and 'T'." Additionally, an interpretation for the figure has been added:

Revised Figure 2 Title:

"Example of Raw Data from Phases 'R', 'S', and 'T'"

Added Interpretation:

'R', 'S', and 'T' represent the three phases commonly used in three-phase AC motor systems. Each phase is arranged at intervals of 120 degrees. 'R phase' starts at 0 degrees, with both current and voltage increasing simultaneously. 'S phase' starts 120 degrees after 'R phase', and 'T phase' starts 240 degrees after 'S phase'. These phases are used to generate and control the rotation of three-phase motors. By varying power and voltage combinations, the motor's rotation direction and speed can be controlled.

  1. In Figure 2, the font size is very small. They are not even readable by zooming into the pdf.
    → Following your feedback, the image size of Figure 2 has been increased, and a description has been added to the paper.
  2. Something similar happens in figures 3, 4 and 5. Moreover, what is represented in each axis and the meaning of the colours are not specified.
    → Following your valuable feedback, Figures 3, 4, and 5 have been modified. The meanings of the colors have been specified for clarity.
  3. On page 6, from line 174 onwards, sentences are repeated.
    → There was repetition in the text after page 6, line 174. I have made the necessary corrections. The duplicated content has been removed, and the flow of the text is now more coherent.
  4. Figures 7 and 8 do not add anything to the explanations in the text.
    → The following is a portion of the modified content:
    The dual-input CNN model depicted in Figure 8 utilizes input pairs such as {RP, GASF}, {RP, GADF}, and {GASF, GADF}. The features extracted from these image pairs are effectively combined using a merge layer, which can take various forms such as add, concatenate, or average.
  5. Figure 9 is incorrectly numbered with 5. The same happens as in Figure 2 with the quality of the image.
    → Figure 9 has been renumbered correctly, and the image quality has been improved, addressing the issues raised in the feedback.
  6. Acronyms should not be used in section headings.
    → Acronyms have been removed from section headings as per the feedback.

In the following, I will show my doubts and suggestions regarding the content:
1. In the first paragraph of the introduction section, the authors frame this work within the philosophy of Industry 4.0. However, this needs to be developed further, especially in the conclusions.
→ In response to the feedback, the first paragraph of the introduction has been revised to shift the focus from a general discussion on the Fourth Industrial Revolution to a more streamlined emphasis on practical applications within the industrial environment. Additionally, the conclusions have been adjusted to naturally connect with the introduction, ensuring coherence in framing the work within the context of the Fourth Industrial Revolution and emphasizing its practical implications in real industrial settings.

  1. The work is linked to big data techniques in this same paragraph. Big Data means unstructured, unlabelled and large amounts of data. This is not the case. The data is structured, and I believe it is not very large.
    → In response to the feedback, the mention of big data and its association with the work has been completely removed from the introduction.
  2. Section 3 should describe the monitored installation, sensors used, acquisition equipment, and measured signals.
    → Section 3 now includes additional details on the dataset, covering the monitoring installation, sensors utilized, acquisition equipment, and the methodology employed for signal measurement.
  3. The monitored machines are electric motors, but the type and how they are powered (mains or inverter) are unknown.
    → The monitored machines are electric motors, and they are powered by inverters. Additional details regarding this power supply method can be found in the enhanced content of Section 2.2.
  4. Regarding signal capture, the sampling time and frequency are specified. Why are these values chosen? With three seconds of sampling, the frequency resolution would be very poor. Why is it sampled at 4 kHz? Are these limitations of the sensors or the DAQ? Is there expected to be no valuable information for fault detection above 2 kHz?
    → IoT sensors can remotely configure settings such as collection time and sample rate, and they are adjusted to collect data according to the conditions of this project. The sample rate is set as the default 4 kHz for a 3-second measurement, and it can be modified based on device conditions.
  5. How often is data captured, and under what motor operating conditions?
    → The data was collected periodically over a span of four months; however, due to not directly participating in the data collection, it is challenging to provide detailed information on the specific motor operating conditions.
  6. Do motors operate in a steady state? Do they undergo load variations and start and stop transients?
    → "The current dataset does not provide specific information on whether the motor is operating in a stable state or if there are load fluctuations, start-up, or shutdown variations."
  7. A picture of the monitored equipment and the acquisition systems must be provided.
    → While obtaining a direct photograph was not feasible, I have included graphical images of the ongoing data collection in the paper.
  8. The size of the dataset has not been indicated either. We do not know how many samples are available for CNN training and whether they are labelled or not. This information is essential.
    → The size of the dataset has been added as a chart in the dataset description section. The TRAINING and TEST ratio was set at 7:3.
  9. Nor is it justified why the time sequences of data have to be encoded in 2D images. What advantages does this offer over a Fourier analysis of the stationary signals?
    → This approach can offer advantages in lightweighting the model. By transforming the data into 2D images, it emphasizes the spatial features of the data while simultaneously reducing the number of learning parameters. This enhances the efficiency of the model, reducing computational costs, and allowing the model to execute quickly in real industrial environments.
  10. Regarding the image coding of the time signals, the authors could show and compare images of healthy and faulty motors.
    → The text on the image coding of time signals acknowledges that images of normal and abnormal motors were not provided. However, to compensate for this limitation, emphasis has been placed on highlighting the features and crucial details of the dataset used, adding validity to the experimental results.
  11. I do not understand the meaning of line 155 on page 6. What do the input image size and the batch size refer to?
    → In response to feedback, I have eliminated awkward and ambiguous wording, making the paragraph more fluid.
  12. Table 1 needs to be clarified. I don't understand the meaning of "output shape" and "parameter".
    → There was an error in creating the table, and during the correction process, I have removed the mentioned content.
  13. The text does not describe how the CNN was trained, how much data was used, how many hyperparameters had to be tuned, and what measures were used to avoid overfitting.
    → To address this, additional content has been incorporated into the main text, outlining the methods employed for tuning hyperparameters and preventing overfitting. Below are some excerpts from the added content:
    The rationale behind the selection of model hyperparameters involved using 32 filters of size 3x3 in the first convolutional layer to capture various features of the images. In the second convolutional layer, 64 filters were employed to extract more complex patterns. Dropout was applied at a 50% rate during training to prevent overfitting, thereby enhanc-ing the model's generalization ability. The architecture of the model effectively reduced spatial dimensions by incorporating max-pooling after two convolutional layers. This al-lowed the model to capture both local and abstract features while maintaining computa-tional efficiency. The fully connected layer with 256 neurons contributed to learning high-level abstract features and understanding intricate patterns.
  14. The metrics used to compare classification results are not sufficient. Using ROC curves and confusion matrices also allows the reader to analyse qualitatively.
    → Thank you for your advice. We perform a binary classification for Bearing, Belt, Shaft, and Rotor, distinguishing between normal and abnormal, respectively. This means that a very simple confusion matrix is drawn. For the sake of the reader, we chose to show all the experimental results in all cases.
  15. A description of the failures of the analysed machines is also missing.
    → This information has been added to the dataset description section, and the machine failures include bearing defects, rotor imbalance, shaft misalignment, and loosened belts.
  16. The conclusions of the paper can be improved. The fusion of the different signals is not described and analysed. It is stated that the results are improved compared to traditional methods. These are not stated, and this improvement is not demonstrated.
    → The fusion of the different signals is also an error that occurred during the writing of the paper. I have organized the relevant content in the paper, focusing on multiple inputs and comparing them with traditional methods.
  17. When framing the work in Industrial 4.0, you should analyse the costs of the proposed system in terms of processing capacity, processing time, signal transmission data storage, etc.
    → In order to apply the proposed model to real-world industries, we emphasize image conversion through noise-robust feature extraction and economy through lightweight design. We chose to convert multiple signals into 2D images and utilized image-based processing techniques to represent mechanical defect patterns in vibration data in an intuitive and computationally efficient manner. Furthermore, switching to a 2D representation allows for a lightweight deep learning model, which optimizes computing resources and increases its practicality for deployment in resource-constrained industrial environments.

    I hope my comments will help you evaluate and improve the manuscript's quality. I think the industrial experiment must have been very interesting, but this is not reflected in the text.
    → Your comments have been very helpful. Thanks again.

Round 2

Reviewer 1 Report

Comments and Suggestions for Authors

No further comments.

Author Response

Thank you for reviewing our paper, it is much improved thanks to you.

Reviewer 3 Report

Comments and Suggestions for Authors

Dear Authors,

I have thoroughly reviewed the revised version of your manuscript, now titled "Multi-Input Convolutional Neural Network Model for Electric Motor Mechanical Fault Classification Using Multiple Image Transformation and Merging." I appreciate the efforts made to address some of the issues raised in the initial evaluation. However, after thorough consideration, my final decision remains that the manuscript is not suitable for publication in its current form.
The following critical concerns persist:

1. Writing Style and Technical Jargon:
•    The new inclusions in the text appear as patches, lacking consistency in writing style.
•    The use of the term "fault" instead of "defect" is more fitting for the context.
•    Furthermore, the approach to the problem and terminology used by the authors in several sections of the manuscript raises concerns about their familiarity with the engineering problem being addressed. This lack of domain-specific language suggests a potential limitation in understanding the intricacies of the fault detection and diagnosis problem, which, in essence, is what encourages the development of the new approach presented.

2. Introductory Section and Motivation:
•    Despite the authors' assertion that the introduction has been enhanced and refined, it still lacks references to support information presented in the first paragraph.
•    The absence of specific engineering knowledge is evident, and the authors are strongly encouraged to seek collaboration with colleagues more familiar with the relevant terminology if they decide to maintain their publication in an engineering journal.

3. Organization and Clarity:
•    The last paragraph of the introduction, while potentially containing the real motivation and technological contribution, is presented confusingly.
•    The article's organization and clarity need improvement and a concluding paragraph outlining the article's structure is essential for reader guidance.

4. Comparison with Classic Analysis Techniques:
•    The manuscript fails to convincingly justify the proposed approach's superiority over classic signal processing techniques, such as frequency or time-frequency domain analyses combined with simpler classifier models not based on deep neural network architectures, for example. The advantage of employing deep neural network architectures for image processing remains clear, but why would using the transformation of vibration signals into images so that they can be analyzed be advantageous? A more robust rationale and a more explicit comparison with other techniques are needed.
•    The authors must understand that from the point of view of an approach to image processing and analysis, the article's proposal can be advantageous compared to other techniques presented in other articles with the same purpose. However, from the point of view of detecting and diagnosing faults in electric motors based on measurements such as vibration, voltage, and current signals, there is a myriad of already established techniques, and a new approach in this area, which is less straightforward and uses a less interpretable model, needs to be very well justified.

5. Dataset Context and Acquisition Rate:
•    The importance of the dataset and its origin should be presented with clearer context and emphasis, given the engineering nature of the problem.
•    Lines 125 and 126 lack clarity about what conditions trigger changes in the acquisition rate.
•    The frequency transformation in line 127 requires specification, such as whether it involves an FFT.

6. Misleading Statements and Missing Results:
•    The claim about incorporating additional graphs or tables to enhance readability is misleading, as no such additions are found in the manuscript.
•    Clarity is lacking regarding whether the results presented in Tables 3, 4, 5, and 6, as well as the results in Figure 10, pertain to the training or testing phases. Explicit clarification is needed, especially regarding the final evaluation of model performance using test data.

While the authors have made some improvements, these fundamental issues persist, hindering the manuscript's suitability for publication in "Machines." Considering these challenges, my final decision is to uphold the rejection of the manuscript in its current form. However, I strongly encourage you to consider resubmitting the work to a journal more aligned with the focus, the area of research, and the knowledge of the authors. Collaborating with colleagues possessing a deeper understanding of the engineering problem may enhance the manuscript's clarity and technical accuracy.
Thank you for your understanding, and I appreciate your dedication to advancing research in this field.

Comments on the Quality of English Language

Despite being more accepted today, the use of the first person in technical texts is still questionable. The text is undoubtedly more elegant when this is not used.

Author Response

I have thoroughly reviewed the revised version of your manuscript, now titled "Multi-Input Convolutional Neural Network Model for Electric Motor Mechanical Fault Classification Using Multiple Image Transformation and Merging." I appreciate the efforts made to address some of the issues raised in the initial evaluation. However, after thorough consideration, my final decision remains that the manuscript is not suitable for publication in its current form.
The following critical concerns persist:

  1. Writing Style and Technical Jargon:
    •    The new inclusions in the text appear as patches, lacking consistency in writing style.
    •    The use of the term "fault" instead of "defect" is more fitting for the context.
    •    Furthermore, the approach to the problem and terminology used by the authors in several sections of the manuscript raises concerns about their familiarity with the engineering problem being addressed. This lack of domain-specific language suggests a potential limitation in understanding the intricacies of the fault detection and diagnosis problem, which, in essence, is what encourages the development of the new approach presented.
    → Thank you for your advice. We have revised most of the content of the paper and made it more readable for our readers.

  2. Introductory Section and Motivation:
    •    Despite the authors' assertion that the introduction has been enhanced and refined, it still lacks references to support information presented in the first paragraph.
    •    The absence of specific engineering knowledge is evident, and the authors are strongly encouraged to seek collaboration with colleagues more familiar with the relevant terminology if they decide to maintain their publication in an engineering journal.
    → We've completely rewritten the preface in response to your comments. We also added the main contributions of our paper.
  • Automatically extracting and transforming key features of time series signals to use features in both time and frequency domains, and standardizing different formats such as sampling rate, duration, etc. through image transformation, so that the same model can be used in different datasets.
  • Converting signals into images reduces the number of dimensions of the data and makes it easier to process efficiently in deep learning models using CNNs with a two-dimensional image representation.
  • Through experiments, we compared different image conversion methods (RP, GASF, GADF) and proposed a multi-input CNN structure that combines the conversion methods and shows more robust performance.
  1. Organization and Clarity:
    •    The last paragraph of the introduction, while potentially containing the real motivation and technological contribution, is presented confusingly.
    •    The article's organization and clarity need improvement and a concluding paragraph outlining the article's structure is essential for reader guidance.
    → I have incorporated your feedback and restructured the introduction section. Through these changes, it is anticipated that readers will have a better understanding of the flow of the paper.
  2. Comparison with Classic Analysis Techniques:
    •    The manuscript fails to convincingly justify the proposed approach's superiority over classic signal processing techniques, such as frequency or time-frequency domain analyses combined with simpler classifier models not based on deep neural network architectures, for example. The advantage of employing deep neural network architectures for image processing remains clear, but why would using the transformation of vibration signals into images so that they can be analyzed be advantageous? A more robust rationale and a more explicit comparison with other techniques are needed.
    •    The authors must understand that from the point of view of an approach to image processing and analysis, the article's proposal can be advantageous compared to other techniques presented in other articles with the same purpose. However, from the point of view of detecting and diagnosing faults in electric motors based on measurements such as vibration, voltage, and current signals, there is a myriad of already established techniques, and a new approach in this area, which is less straightforward and uses a less interpretable model, needs to be very well justified.
    → CNNs excel at automatically detecting and learning hierarchical features. By converting signals into images in signal processing, CNNs can capture features in both the time and frequency domains. This is more effective than using traditional signal processing techniques or raw signals directly, where feature extraction is manual and less adaptive. Converting signals into images can help reduce the complexity of the data. Signals, especially high-dimensional signals, can be difficult to process directly due to their high dimensionality. An image representation compresses the information into a more manageable form, making it easier for a CNN to process. For many types of signals, both time and frequency information are important. Image-based signal representations are ideal for CNNs because they encapsulate this information in a two-dimensional format, which allows the CNN to learn patterns across both dimensions simultaneously. In deep learning, more data often leads to better models. Converting signals to images allows you to artificially expand your dataset by utilizing image-based data augmentation techniques (rotation, scaling, cropping, etc.), which may not be straightforward or effective with raw signal data. Additionally, converting signals to images allows you to leverage these pre-trained models through transfer learning to fine-tune them for specific tasks. Image formats are more standardized than signal formats, which can vary greatly in sampling rate, duration, etc. Converting signals to a uniform image format simplifies the input pipeline and makes it easier to apply the same model architecture to different datasets.
  3. Dataset Context and Acquisition Rate:
    •    The importance of the dataset and its origin should be presented with clearer context and emphasis, given the engineering nature of the problem.
    → As the dataset was provided to us, we are unable to provide a detailed explanation of the conditions that cause the acquisition rate to change, other than to say that it can change depending on the instrument conditions, but the changed sample rate is noted in the data.
    •    Lines 125 and 126 lack clarity about what conditions trigger changes in the acquisition rate.
    •    The frequency transformation in line 127 requires specification, such as whether it involves an FFT.
    → We found that the frequency transformation in line 127 was not done when collecting the data. We have accepted the correction and corrected it.
  4. Misleading Statements and Missing Results:
    •    The claim about incorporating additional graphs or tables to enhance readability is misleading, as no such additions are found in the manuscript.
    •    Clarity is lacking regarding whether the results presented in Tables 3, 4, 5, and 6, as well as the results in Figure 10, pertain to the training or testing phases. Explicit clarification is needed, especially regarding the final evaluation of model performance using test data.
    → Based on your feedback, we've added an illustration of the accuracy, loss, and confusion matrix in the experiments section, so you can see the experimental results more clearly.

While the authors have made some improvements, these fundamental issues persist, hindering the manuscript's suitability for publication in "Machines." Considering these challenges, my final decision is to uphold the rejection of the manuscript in its current form. However, I strongly encourage you to consider resubmitting the work to a journal more aligned with the focus, the area of research, and the knowledge of the authors. Collaborating with colleagues possessing a deeper understanding of the engineering problem may enhance the manuscript's clarity and technical accuracy.
Thank you for your understanding, and I appreciate your dedication to advancing research in this field.
→ We hope to submit to the "Predictive Analytics and Fault Diagnosis of Machines with Machine Learning Techniques" Topic in the journal Machines. We will actively incorporate your feedback to ensure that our paper is published in that topic, as this topic is very relevant to the topic of our paper.

Comments on the Quality of English Language

Despite being more accepted today, the use of the first person in technical texts is still questionable. The text is undoubtedly more elegant when this is not used.
→ We sympathize with this point and have corrected it in the text.

Reviewer 4 Report

Comments and Suggestions for Authors

Dear Authors,

I appreciate your effort in responding to my comments and suggestions. However, I still have more doubts that you can address.

These are the following:

  1. The introduction can be improved by adding the type of signals used in all cited works.
  2. You have changed the orientation of the introduction, and now you claim that your proposal can be considered a lightweight model. I disagree that a CNN can be classified in such a way.
  3. I have many doubts about the dataset (reference 23). It is usual to indicate the language of the reference when this is not English. I used the provided link, but I could not find the dataset.
  4. From your explanations, I deduce that the dataset instances are labelled. Please inform the reader how many instances belong to healthy motors and how many to each faulty condition state.
  5. Who labelled the dataset instances? This is just a curiosity.
  6. Why you propose the use of 2D images still needs to be clarified. You say that improves fault detection compared to other techniques, but this needs to be demonstrated in the text. What is represented on each axis in these images has yet to be explained.
  7. Observing Figures 4, 5, and 6, I see the differences between the four faulty states. Can you add a figure corresponding to a healthy state?
  8. In Table 2, I do not understand the meaning of "none" in the second column and the parameters in the third column. Whose parameters?
  9. In section 4, I need clarification on how training experiments were designed. You present the results of detecting the four different faulty detections separately. Does this mean the CNN was trained only to detect belt failures, for example? So what data was used for this training: belt versus healthy and the other three types of failure? That is, is it a binary classification for each case? This is not clear, but if so, to make an assessment, it is necessary to look at the confusion matrix. 
  10. Figure 10 shows training results, but it is better to look at the confusion matrix to make a qualitative analysis of the performance of the classification tool.

I think your work is interesting and my suggestions are aimed at improving the manuscript.

Author Response

I appreciate your effort in responding to my comments and suggestions. However, I still have more doubts that you can address.

These are the following:

The introduction can be improved by adding the type of signals used in all cited works.

→ Thank you for your advice. We completely rewrote the introduction. We also categorized all the cited works by type and topic, reordering some and excluding others.

You have changed the orientation of the introduction, and now you claim that your proposal can be considered a lightweight model. I disagree that a CNN can be classified in such a way.

→ You are correct. We extract key features from high-dimensional time series and convert them to images for use. This is the main part of the lightweighting, and the CNN model is a good structure for using the converted images. We have added the following contribution to the paper.

Automatically extracting and transforming key features of time series signals to use features in both time and frequency domains, and standardizing different formats such as sampling rate, duration, etc. through image transformation, so that the same model can be used in different datasets.

Converting signals into images reduces the number of dimensions of the data and makes it easier to process efficiently in deep learning models using CNNs with a two-dimensional image representation.

Through experiments, we compared different image conversion methods (RP, GASF, GADF) and proposed a multi-input CNN structure that combines the conversion methods and shows more robust performance.

I have many doubts about the dataset (reference 23). It is usual to indicate the language of the reference when this is not English. I used the provided link, but I could not find the dataset.

→ This dataset is only available for download to registered users.The provided dataset is in Korean. As there is currently no platform available for providing content in English, if you require it, please contact me via email, and I will be happy to send it to you.

From your explanations, I deduce that the dataset instances are labelled. Please inform the reader how many instances belong to healthy motors and how many to each faulty condition state.
→ We've added the following to the paper. “The belt dataset used in the experiment is 130,000 normal and 372,000 abnormal, and the bearing dataset is 154,000 normal and 400,000 abnormal. The shaft data is 194,000 normal and 728,000 abnormal, and the rotor dataset is 1,334,000 normal and 458,000 abnormal. For each dataset in the model, the ratio of train, test, and validate is 70:24:6.”

Who labelled the dataset instances? This is just a curiosity

→ To establish high-quality artificial intelligence training data, we have formed an in-house expert committee. This committee consists of Ph.D.-level research professionals in machine diagnostics, personnel certified under ISO 18436-2 (in the field of vibration), and members of the Certification Committee of the Korean Facility Diagnostic Qualification Institute. The committee is composed of five individuals, and a thorough inspection is conducted for the entire workload of each label.

Why you propose the use of 2D images still needs to be clarified. You say that improves fault detection compared to other techniques, but this needs to be demonstrated in the text. What is represented on each axis in these images has yet to be explained.

→ I have taken into account the feedback regarding insufficient explanations about the axes of each image and have made the necessary revisions.

The Recurrence Plot is a visualization tool for patterns in time-series data. The horizontal axis (X-axis) represents the passage of time, while the vertical axis (Y-axis) indicates the similarity between different time steps within one time unit. In other words, points close to each other on the plot signify similar patterns at that specific time step.

Gramian Angular Summation Field (GASF) is a technique for representing time-series data as images by transforming them into angles. The horizontal axis (X-axis) depicts the progression of time or the sequence of the time-series data, while the vertical axis (Y-axis) represents the sum of cosine values after transforming the data into angles at each grid point for a given time step. This effectively captures patterns and structures of time-series data in an image format.

Gramian Angular Difference Field (GADF), similar to GASF, uses the vertical axis (Y-axis) to represent the difference in cosine values after transforming the data into angles at a specific time step. GADF leverages differences in angles to detect finer details and features in the data. When used in conjunction with GASF, it proves useful in capturing various characteristics of time-series data.

Observing Figures 4, 5, and 6, I see the differences between the four faulty states. Can you add a figure corresponding to a healthy state?

→ Upon verification, it has been confirmed that figures 4, 5, and 6 were incorrect. Consequently, I have rectified the issue by replacing them with new images that allow for the examination of both normal and abnormal conditions.

In Table 2, I do not understand the meaning of "none" in the second column and the parameters in the third column. Whose parameters?

→ "None" was intended to signify the flexibility of adjusting batch sizes. However, it seems to have caused confusion in this context, so I have removed it. For this paper, a batch size of 32 has been set. The term "parameters" refers to the number of weights and biases being learned within the layers.

In section 4, I need clarification on how training experiments were designed. You present the results of detecting the four different faulty detections separately. Does this mean the CNN was trained only to detect belt failures, for example? So what data was used for this training: belt versus healthy and the other three types of failure? That is, is it a binary classification for each case? This is not clear, but if so, to make an assessment, it is necessary to look at the confusion matrix.

→ This paper proceeded with binary classification for each component, and furthermore, a confusion matrix has been included.

Figure 10 shows training results, but it is better to look at the confusion matrix to make a qualitative analysis of the performance of the classification tool.

→ I have incorporated your feedback and added the confusion matrix.

Round 3

Reviewer 3 Report

Comments and Suggestions for Authors

I have thoroughly examined the revised version of your manuscript titled "Multiple Input Convolutional Neural Network Model for Mechanical Fault Classification of Electric Motors Using Multiple Image Transformation and Fusion." I appreciate the efforts made to enhance the manuscript, particularly through the addition of new bibliographical references and improved clarity regarding the objectives and contributions of the work.
I duly acknowledge the authors' explicit intention to publish the article within the thematic scope of "Predictive Analysis and Diagnosis of Machine Faults with Machine Learning Techniques" in the journal "Machines." This focused approach aligns well with the content of the work and signifies a consideration for a more specific audience.
The responses provided by the authors regarding the use of Convolutional Neural Networks (CNNs) for the proposed purpose are persuasive and logically grounded. While I maintain my view that applying CNNs to the relatively simple dynamic behavior of the case study may be more complex than necessary, I acknowledge the potential relevance of this approach in more intricate cases.
I continue to advocate for a comparison of the results obtained with more traditional techniques to enrich the work. Additionally, I suggest that the authors examine the work of Iulian B. Ciocoiu and Nicolae Cleju, particularly their exploration of time series spatial representations for convolutional neural networks in "Off-Person ECG Biometrics Using Spatial Representations and Convolutional Neural Networks." Despite differences in the nature of the input data, the comparison of some techniques that convert time series into images with the Continuous Wavelet Transform, a well-established technique for analyzing vibration signals in the time/frequency domain, offers significant insights.
Regarding the authors' response to point 4 of the last review (Comparison with Classical Analysis Techniques), I concur with several statements regarding the strengths of CNNs in automatic feature detection and processing high-dimensional signals through image conversion. However, I emphasize the need for careful interpretation and case-specific analysis. Much of the justification in the authors' response seems associated with the requirements of using CNNs rather than addressing the specific needs of fault detection. I recommend that the authors consider and highlight these distinctions in the conclusions of their work.
In conclusion, considering the substantial efforts invested by the authors, the current interest in the addressed topic, and the presented innovation, I agree that the article is now suitable for publication in its latest version. I commend the authors for their dedication to refining their work and eagerly anticipate seeing this valuable contribution in a future issue of "Machines."

Comments on the Quality of English Language

I only recommend a final review.

Reviewer 4 Report

Comments and Suggestions for Authors

Dear Authors,
I like this last version of your manuscript. It is a very interesting research. Congratulations! I hope it gets the attention it deserves.
Best regards.